# Mitigating Multimodal Hallucinations via Gradient-based Self-Reflection

## Abstract

Multimodal large language models achieve strong performance across diverse tasks but remain prone to hallucinations, where outputs are not grounded in visual inputs. This issue can be attributed to two main biases: text–visual bias, the overreliance on prompts and prior outputs, and co-occurrence bias, spurious correlations between frequently paired objects. We propose Gradient-based Influence-Aware Constrained Decoding (GACD), an inference-based method, that addresses both biases without auxiliary models, and is readily applicable to existing models without finetuning. The core of our approach is bias estimation, which uses first-order Taylor gradients to understand the contribution of individual tokens—visual features and text tokens—to the current output. Based on this analysis, GACD mitigates hallucinations through two components: (1) suppressing spurious visual features correlated with the output objects, and (2) rebalancing cross-modal contributions by strengthening visual features relative to text. Experiments across multiple benchmarks demonstrate that GACD effectively reduces hallucinations and improves the visual grounding of MLLM outputs.

## 1 Introduction

Recent advances in Multimodal Large Language Models (MLLMs) have demonstrated remarkable capabilities in producing coherent and context-aware content across a wide range of domains (Bai et al., 2023; Dai et al., 2023; Chen et al., 2024b; Liu et al., 2024a; Ye et al., 2024). Despite their impressive advancements, these models remain prone to hallucination, wherein the generated text is not faithfully grounded in the visual modality (Rohrbach et al., 2018; Li et al., 2023b). This limitation poses a critical barrier to establishing trust in the outputs of MLLMs.

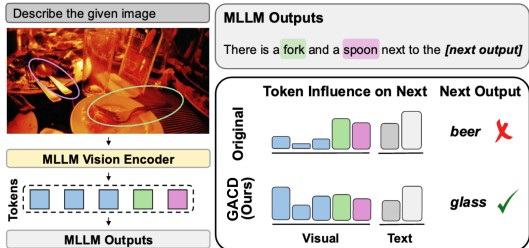

Figure 1: Overview of our influence-aware constrained decoding framework, which mitigates hallucinations by regulating token-level influence. It reduces text–visual bias by enhancing visual token influence (blue bars) in alignment with the most influential text inputs— prompts (gray) or previous outputs (white). It further mitigates co-occurrence bias through anchor-specific suppression, selectively suppressing visual tokens (green, magenta) anchored to previously emitted nouns.

The hallucinations observed in MLLMs can be largely attributed to two fundamental biases (Kang & Choi, 2023; Li et al., 2023b; Kim et al., 2024). **Text-visual bias** refers to the excessive reliance on textual information—such as the input prompt and previously generated outputs—while neglecting the visual modality during generation. This bias becomes particularly pronounced in longer sequences, where MLLMs tend to depend more heavily on prior text and increasingly disregard visual cues (Zhou et al., 2023b; Favero et al., 2024). **Co-occurrence bias** arises from spurious statistical correlations embedded in the training data, which lead models to erroneously predict the presence of non-existent objects based on their frequent co-occurrence with observed objects in the visual inputs (Li et al., 2023b). This bias is particularly challenging to mitigate, and existing approaches largely rely on statistical priors rather than offering statistically agnostic solutions (Kang & Choi, 2023; Zhou et al., 2023a).

Existing efforts to mitigate hallucinations in MLLMs can be broadly categorized into inference-based methods, which operate at the decoding stage (Chen et al., 2024a; Favero et al., 2024; Leng et al., 2024; Park et al., 2024; Woo et al., 2024), and training-based approaches, which intervene during model optimization (Ben-Kish et al., 2023; Chen et al., 2023b; Kang & Choi, 2023; Sun et al., 2023; Jiang et al., 2024; Yue et al., 2024). Inference-based approaches are valued for their cost-effectiveness, as they avoid the need for additional data collection, data bias examination, or extensive model retraining. However, these methods offer limited insight into the severity of underlying biases, leaving the root causes of hallucination insufficiently understood. In addition, some inference-based methods rely on auxiliary models—such as segmentation networks Chen et al. (2024a), detection systems Kan et al. (2024) , or even additional MLLMs (Radford et al., 2021; Deng et al., 2024; Xing et al., 2024)—which undermine their purported cost-effectiveness.

Another limitation of existing methods lies in their lack of granularity when adjusting the underlying biases in MLLMs. Most approaches rely on heuristically tuned priors, which vary across datasets and fail to generalize reliably Leng et al. (2024); Zhao et al. (2024). Moreover, they apply uniform weighting across all visual features, offering no mechanism to selectively adjust bias at the level of individual featuresZhang et al. (2024b); Manevich & Tsarfaty (2024). This coarse treatment limits their effectiveness in mitigating co-occurrence bias, which arises from spurious statistical correlations between objects that are often represented by distinct visual features.

In this work, we propose an inference-based method that simultaneously addresses both text–visual bias and co-occurrence bias, without relying on auxiliary models or external supervision. The core of our approach is the estimation of underlying bias, achieved by quantifying the contribution of individual tokens—both visual features and text tokens—through gradients derived from a first-order Taylor expansion. Building on this analysis, the method mitigates hallucinations by reweighting tokens via two key components: (1) suppressing the influence of visual features that exhibit strong spurious correlations with the current output token, thereby reducing co-occurrence bias; and (2) rebalancing cross-modal contributions by enhancing the role of visual features to align more closely with that of text tokens in generating the current output. As illustrated in Fig. 1, our method, GACD, corrects hallucinated predictions—such as the spurious generation of "beer" in the presence of "fork" and "spoon"—by amplifying the contributions of visual tokens unrelated to those nouns, leading to outputs that are more faithfully grounded in the visual modality. Note also that our method is readily applicable to existing MLLMs at inference time.

We summarize our main contributions as follows.

- We introduce an inference-based method for hallucination mitigation in MLLMs, built on a principled estimation of underlying bias via gradients obtained from a first-order Taylor expansion. This estimation provides a mechanism for understanding and granularly adjusting their influences of individual visual features and text tokens on the generation of the current output token, all without requiring auxiliary models or finetuning.

- We design two complementary modules: (i) suppression of spurious visual features correlated with the current output token to alleviate co-occurrence bias, and (ii) cross-modal rebalancing to enhance the contributions of visual features relative to text tokens, thereby addressing text–visual bias.

- Extensive experiments demonstrate that GACD mitigates hallucinations and enhances accuracy without sacrificing information. GACD achieves up to $8\%$ increase in overall score on AMBER Wang et al. (2023), an $8\%$ F1 boost on POPE Li et al. (2023b), up to $45\%$ improvement in detailness and a $92\%$ accuracy gain on LLaVA-QA90 Liu et al. (2024b).

## 2 RELATED WORK

**Hallucination and Bias**. Hallucinations in LLMs often arise from biases in the training data McKenna et al. (2023); Huang et al. (2025), while in MLLMs, studies Tonmoy et al. (2024); Li et al. (2023b); Fu et al. (2024) show that hallucinations are closely linked to biases like text-visual and co-occurrence biases. Additionally, biases related to output position, which increase the risk of hallucination as output length grows, have been examined in Favero et al. (2024); Zhou et al. (2023b). Existing methods Li et al. (2023b); Fu et al. (2024); Kim et al. (2024) typically report only overall statistics, lacking a mathematical, sample-wise bias measurement. This distinction is impor-

tant, as biases can vary case by case. Our approach measures sample-dependent bias via token-level gradient sensitivities, revealing how pre-trained MLLM parameters embed these biases Kim et al. (2019); Guo et al. (2024), and enabling self-reflective hallucination mitigation.

**Hallucination Mitigation**. Training-related hallucination mitigation methods Chen et al. (2023a); Jiang et al. (2024); Yue et al. (2024); Peng et al. (2025); Zadeh et al. (2025) are expensive, requiring access to training data and specialized statistical analysis. Among them, LPOI Zadeh et al. (2025) also employs an object-aware framework, highlighting the effectiveness of modeling object-level information for mitigating hallucinations. Reinforcement-learning approaches Xing et al. (2024); Deng et al. (2024); Zhai et al. (2024) rely on supplementary feedback, often from human annotators or auxiliary LLMs/MLLMs, and the latter may themselves hallucinate. By contrast, post-decoding techniques modify model logits at inference time without further training or external feedback, making them lightweight add-ons. In text-only LLMs, such methods aim to align outputs with factual knowledge Chuang et al. (2023); Li et al. (2023a). In MLLMs, post-decoding strategies emphasize the role of visual inputs Leng et al. (2024); Zhao et al. (2024); Favero et al. (2024) and can be classified into image-level and token-level interventions. Image-level decoding methods Zhang et al. (2024b); Manevich & Tsarfaty (2024) treat all objects in the input image uniformly, limiting their effectiveness in addressing co-occurrence hallucinations. Existing token-level methods either rely on external segmentation Chen et al. (2024a) and detection models Kan et al. (2024) or lack awareness of object-related decoupling Woo et al. (2024). Moreover, these methods typically introduce an implicit trade-off between accuracy and informativeness, reducing hallucinations at the expense of omitting valid details. Attention-based methods Tang et al. (2025); Zhang et al. (2024a) require careful selection of specific layers and often introduce model-specific adjustments or heuristics. In contrast, our GACD directly estimates embedded bias and decouples object-aware visual tokens, enabling sample-specific hallucination mitigation without external data, models, or model-specific adjustments, while achieving a more favorable balance between accuracy and informativeness.

## 3 METHOD

In this section, we provide background on MLLMs, introduce the concept of token influence, and explain how GACD balances token influence to mitigate hallucinations.

### 3.1 BACKGROUND ON MLLMS

MLLMs generate a finite token sequence $\mathbf{y} = [y_1, \ldots, y_M]$ in response to a visual input (image or video) and a textual prompt. Let $\mathcal{V}$ be a finite vocabulary. The prompt is tokenized as $\mathbf{t}^p = [t_1^p, \ldots, t_N^p]$ with $t_n^p \in \mathcal{V}$. The visual input is encoded by a visual encoder into features, which are then projected into the token-embedding space $\mathbb{R}^d$, yielding visual tokens $\mathbf{t}^v = [t_1^v, \ldots, t_S^v]$ with $t_s^v \in \mathbb{R}^d$, where $d$ is the shared token embedding dimension used for $\mathcal{V}$.

A MLLM with parameters $\theta$ computes, at each decoding step $m$, a logit vector

$$\mathbf{z}_m = \pi_\theta(\mathbf{t}^v, \mathbf{t}^p, \mathbf{y}_{<m}) \in \mathbb{R}^{|\mathcal{V}|}, \qquad \mathbf{y}_{<m} = [y_1, \ldots, y_{m-1}] \text{ (empty when } m = 1). \quad (1)$$

This induces a categorical next-token distribution via the softmax $\sigma : \mathbb{R}^{|\mathcal{V}|} \to \Delta^{|\mathcal{V}|-1}$:

$$p_\theta(y_m \mid \mathbf{t}^v, \mathbf{t}^p, \mathbf{y}_{<m}) = \left[ \sigma(\mathbf{z}_m) \right]_{y_m}, \qquad 1 \le m \le M, \quad (2)$$

where $\sigma(\mathbf{z}_m) \in \Delta^{|\mathcal{V}|-1}$ denotes the probability distribution[1] over the vocabulary, and $[\cdot]_{y_m}$ selects the component corresponding to token $y_m \in \mathcal{V}$. At inference, $y_m$ is sampled from this categorical distribution (e.g., greedy, beam search). The sequence likelihood factorizes by the chain rule:

$$p_\theta(\mathbf{y} \mid \mathbf{t}^v, \mathbf{t}^p) = \prod_{m=1}^{M} p_\theta(y_m \mid \mathbf{t}^v, \mathbf{t}^p, \mathbf{y}_{<m}). \quad (3)$$

Given a dataset $\mathcal{D}$ of $(\mathbf{t}^v, \mathbf{t}^p, \mathbf{y})$, maximum-likelihood training (or fine-tuning) estimates $\theta^\star$ by maximizing the conditional log-likelihood. Pretrained MLLMs encode statistical regularities (including spurious correlations) from training data in $\theta^\star$; such behavior can be probed without changing $\theta^\star$ via parameter-dependent analyses (e.g., gradients/attributions or counterfactual decodings) Kim et al. (2019); Guo et al. (2024), enabling self-reflective bias interpretation.

---

[1]We use "confidence" to denote the model-assigned probability of the emitted token.

## 3.2 GRADIENT-BASED TOKEN INFLUENCE ESTIMATION

To capture these embedded biases, we measure how each input token perturbs the output logits. Let $\mathbf{z}_m^\star \in \mathbb{R}^{|\mathcal{V}|}$ denote the step-$m$ logits $\mathbf{z}_m^\star = \pi_{\theta^\star}(\mathbf{t}^v, \mathbf{t}^p, \mathbf{y}_{<m})$. Around a reference sample point $(\mathbf{t}^{v(0)}, \mathbf{t}^{p(0)}, \mathbf{y}_{<m}^{(0)})$, the first-order Taylor expansion Spivak (1980) of the logits $\mathbf{z}_m^\star$ is

$$\mathbf{z}_m^\star \approx \sum_{s=1}^{S} \mathbf{g}_{ms}^v t_s^v + \sum_{n=1}^{N} \mathbf{g}_{mn}^p t_n^p + \sum_{i=1}^{m-1} \mathbf{g}_{mi}^y y_i + Const, \qquad (4)$$

where $Const$ denotes other terms that are constant w.r.t., $\mathbf{t}^v$ and $\mathbf{t}^p$ and the token-wise Jacobians are

$$\mathbf{g}_{ms}^v := \left.\frac{\partial \mathbf{z}_m^\star}{\partial \mathbf{t}_s^v}\right|_{\mathbf{t}^v=\mathbf{t}^{v(0)}}, \qquad \mathbf{g}_{mn}^p := \left.\frac{\partial \mathbf{z}_m^\star}{\partial \mathbf{t}_n^p}\right|_{\mathbf{t}^p=\mathbf{t}^{p(0)}}, \qquad \mathbf{g}_{mi}^y := \left.\frac{\partial \mathbf{z}_m^\star}{\partial \mathbf{y}_i}\right|_{\mathbf{y}=\mathbf{y}_{<m}^{(0)}}, \qquad (5)$$

where $|.$ indicate evaluation at the reference sample point. Taylor expansion details are in supplementary Sec. A. Each $\mathbf{g}_{ms}^v$, $\mathbf{g}_{mn}^p$, $\mathbf{g}_{mi}^y$ indicate a small token perturbation in its corresponding embedding space to a perturbation of the predict logit vector in $\mathbb{R}^{|\mathcal{V}|}$. We score the importance of each input token by the Manhattan norm of its gradient:

$$I_{ms}^v = \|\mathbf{g}_{ms}^v\|_1, \qquad I_{mn}^p = \|\mathbf{g}_{mn}^p\|_1, \qquad I_{mi}^y = \|\mathbf{g}_{mi}^y\|_1, \qquad (6)$$

and $I_{ms}^v[c]$ represents the gradient from the output vocabulary $c$ with respect to each visual tokens. Aggregating over tokens yields step-$m$ group-level influences:

$$\mathtt{I}_m^v = \sum_{s=1}^{S} I_{ms}^v, \qquad \mathtt{I}_m^p = \sum_{n=1}^{N} I_{mn}^p, \qquad \mathtt{I}_m^y = \sum_{i=1}^{m-1} I_{mi}^y. \qquad (7)$$

These quantities decompose, at the sample level, how visual tokens, prompt tokens, and prior outputs contribute to the logit of $y_m$, enabling interpretation of bias per sample.

## 3.3 INFLUENCE-AWARE CONSTRAINED DECODING

GACD builds on token influence estimation with two components: (i) Object-aware Visual Token Grouping and (ii) Anchor-specific Influence-weighted Decoding. At step $m$, the former partitions visual tokens into object-related $\mathbf{t}^o$ and unrelated $\mathbf{t}^u$ based on objects detected in $\mathbf{y}_{<m}$. The latter extends contrastive decoding Li et al. (2022) by forming *Anchor-specific* negative guidance logits from pre-mentioned objects and computing a decoding weight $\alpha_m$ from token-influence measurements.

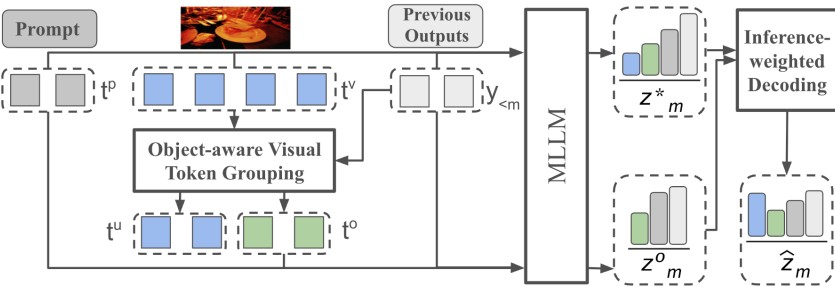

Figure 2: Overview of GACD. The method comprises (i) Object-aware Visual Token Grouping and (ii) Anchor-specific Influence-Weighted Decoding. At step $m$, previously mentioned objects are detected from $\mathbf{y}_{<m}$; visual tokens are partitioned into object-related textcolordarkgreen$\mathbf{t}^o$ and unrelated $\mathbf{t}^u$ via token influence (Sec. 3.2). Anchor-specific Influence-weighted Decoding extends contrastive decoding with token influence, explicitly amplifying the influence of $\mathbf{t}^u$ to jointly counter text-visual and co-occurrence biases; negative-guidance logits $\mathbf{z}_m^o$ are generated from $\{\mathbf{t}^o, \mathbf{t}^p, \mathbf{y}_{<m}\}$ to suppress text tokens and anchor-specific visual cues. Grouping is invoked only for noun prediction (where co-occurrence arises between object pairs); for non-noun prediction, we set $\mathbf{t}^o = \varnothing$ and uniformly amplify all visual tokens to balance text–visual bias.

**Object-aware Visual Token Grouping.** For each step $m$, we detect nouns in $\mathbf{y}_{<m}$ and treat each noun $y_i$ as an object mention. To link a mention to visual evidence, we measure the influence $I_{is}^v$ of visual token $s$ on step $i$. For every noun $y_i$, the visual token with maximal influence is selected to form a mask $\mathcal{M}_{is}$. The cumulative object mask at step $m$ aggregates all prior noun-linked tokens:

$$\mathcal{M}_{ms} = \mathbf{1}\left[\sum_{i=1}^{m-1} \mathcal{M}_{is} > 0\right], \quad \mathcal{M}_{is} = \mathbf{1}\left[y_i \in \text{Noun} \ \wedge \ s = \arg\max_j I_{ij}^v\right], \qquad (8)$$

where $\mathbf{1}[\cdot]$ is the indicator and $\wedge$ is logical AND.

The mask $\mathcal{M}_{ms}$ identifies visual tokens linked to nouns emitted before $m$. We then partition the visual tokens into *object-related ($\mathbf{t}^o$)* and *unrelated-to-objects ($\mathbf{t}^u$)* sets via a Hadamard product:

$$\mathbf{t}^o = \mathbf{t}^v \odot \mathcal{M}_m, \qquad \mathbf{t}^u = \mathbf{t}^v \odot (\mathbf{1} - \mathcal{M}_m). \qquad (9)$$

Object-related and unrelated influences at step $m$ are

$$\mathtt{I}_m^o = \sum_{s=1}^{S} \|\mathbf{g}_{ms}^v\|_1 \, \mathcal{M}_{ms}, \qquad \mathtt{I}_m^u = \sum_{s=1}^{S} \|\mathbf{g}_{ms}^v\|_1 \, (1 - \mathcal{M}_{ms}). \qquad (10)$$

Masking and grouping are applied only during noun prediction (mitigate co-occurrence hallucination from object pairs). For non-noun steps, all elements in $\mathcal{M}_m$ are set to 0, yielding an empty $\mathbf{t}^o$.

**Anchor-specific Influence-weighted Decoding.** Let $\mathbf{z}_m^o = \pi_{\theta^\star}(\mathbf{t}^o, \mathbf{t}^p, \mathbf{y}_{<m})$ the *anchor-specific* negative logits and $\mathbf{z}_m^\star = \pi_{\theta^\star}(\mathbf{t}^v, \mathbf{t}^p, \mathbf{y}_{<m})$ be the *original* logits. We adjust logits by

$$\hat{\mathbf{z}}_m = (1 + \alpha_m)\, \mathbf{z}_m^\star - \alpha_m\, \mathbf{z}_m^o, \qquad (11)$$

with $\alpha_m \geq 0$. In the probability space, moving along $\mathbf{z}_m^\star - \mathbf{z}_m^o$ increases the KL divergence $D_{\mathrm{KL}}(\sigma(\mathbf{z}_m^\star)\|\sigma(\mathbf{z}_m^o))$ (see Sec. B). The original logits distribution $\mathbf{z}_m^\star$ can be viewed as $\pi_{\theta^\star}(\mathbf{t}^u, \mathbf{t}^o, \mathbf{t}^p, \mathbf{y}_{<m})$, i.e., a joint distribution that additionally depends on $\mathbf{t}^u$ compare to $\mathbf{z}_m^o$. Increasing the KL divergence therefore emphasizes the contribution of tokens $\mathbf{t}^u$, which are unrelated to previous mentioned objects, thereby mitigating co-occurrence bias in noun prediction. For non-noun steps, $\mathbf{t}^u$ coincides with $\mathbf{t}^v$, meaning that all visual tokens are emphasized. This adjustment helps reduce text–visual hallucination.

When analyzing token influence of $\hat{\mathbf{z}}_m$ in Eq. 11, the chain rule shows that $\mathbf{t}^u$ occur only in the original logits $\mathbf{z}_m^\star$ and are amplified by $(1 + \alpha_m)$, whereas other inputs also contribute to $\mathbf{z}_m^o$ and therefore undergo smaller influence changes. Let $\tilde{\mathtt{I}}_m^o, \tilde{\mathtt{I}}_m^p, \tilde{\mathtt{I}}_m^y$ denote group influences computed on the negative branch $\mathbf{z}_m^o$ (analogous to equation 7). We then choose $\alpha_m$ so that the influence of $\mathbf{t}^u$ matches the *dominant text* level, $\mathtt{I}_m^t := \max(\mathtt{I}_m^p, \mathtt{I}_m^y)$. Aligning $\mathbf{t}^u$ influence with the question prompt $\mathtt{I}_m^p$ is crucial for visually grounded responses, while balancing with previous outputs $\mathtt{I}_m^y$ prevents visual forgetting.

$$\alpha_m = \frac{\mathtt{I}_m^t - \mathtt{I}_m^v}{\mathtt{I}_m^v - \tilde{\mathtt{I}}_m^o + \tilde{\mathtt{I}}_m^t - \mathtt{I}_m^t}, \tilde{\mathtt{I}}_m^t = \begin{cases} \tilde{\mathtt{I}}_m^p & \text{if } \mathtt{I}_m^p \geq \mathtt{I}_m^y \\ \tilde{\mathtt{I}}_m^y & \text{otherwise} \end{cases} \qquad (12)$$

Unlike existing decoding methods Zhou et al. (2023b); Favero et al. (2024); Leng et al. (2024), which rely on adaptive plausibility constraints (e.g., prediction confidence) and require experimental tuning to determine optimal thresholds, our approach explicitly enforces non-negativity on the influence of object-related visual and prompt tokens. This corresponds to the following upper-bound condition:

$$0 \leq \alpha_m \leq \min\left\{\frac{\mathtt{I}_m^o}{\tilde{\mathtt{I}}_m^o - \mathtt{I}_m^o}, \frac{\mathtt{I}_m^p}{\tilde{\mathtt{I}}_m^p - \mathtt{I}_m^p}\right\}. \qquad (13)$$

**Sample-dependent early stopping.** Additionally, since hallucinations are more likely in long generations Zhou et al. (2023b); Peng et al. (2025), we introduce a sample-dependent stopping criterion based on visual influence. Specifically, if the visual influence ratio $r_m^v$ of the token following the end-of-sequence (EOS) falls below a threshold $\epsilon$,

$$r_m^v := \frac{\mathtt{I}_m^v}{\mathtt{I}_m^v + \mathtt{I}_m^p + \mathtt{I}_m^y} < \epsilon \quad \text{and} \quad y_{m-1} = \text{EOS}. \qquad (14)$$

Early stopping is triggered to prevent further output generation with minimal visual grounding.

## 4 EXPERIMENTS

The proposed method is evaluated for both the open-ended generative tasks and the discriminative tasks. We use Amber Wang et al. (2023), MSCOCO Lin et al. (2014) and LLaVa-QA90 Liu et al. (2024b) datasets for the generative task, and Amber Wang et al. (2023) and POPE Li et al. (2023b) datasets on the discriminative tasks.

**Evaluation Metrics**. For generative image captioning, we focus on object hallucination and follow Deng et al. (2024) report the Caption Hallucination Assessment with Image Relevance (CHAIR) Rohrbach et al. (2018) score, which includes sentence-level ($hal$,$C_S$) and instance-level ($cha$,$C_I$) percentages, instance-level recall ($R$,$cov$), and the average generated length ($Len$) [2], as well as co-occurrence object hallucination ($cog$) and the overall $score$ as suggested by Wang et al. (2023). For generative VQA, follow Leng et al. (2024); Huang et al. (2024) GPT-4V Achiam et al. (2023) is used to score both accuracy (Acc) and detailedness (Det) on a scale of 10. For discriminative tasks, hallucination manifests as a 'yes/no' misclassification we report both accuracy and F1 score.

**Implementation Details**. The maximum output length is set to 256 across all models, with other model parameters kept at their defaults. To prevent excessive modifications, we set the maximum amplification factor, $\alpha_m$, to 5 for discriminative tasks and 3 for generative tasks. We empirically set the early stopping thresholds $\epsilon$ as follows: LLaVA-v1.5 and LLaVA-v1.6: 7%, InstrucBLIP: 25%, mPLUG-Owl2: 2.5%, and InternVL2: 10%. All experiments are performed on an NVIDIA A40 GPU with batch size of 1. Unless otherwise specified, we use greedy sampling Graves (2013).

### 4.1 RESULTS ON OPEN-ENDED GENERATION

In this section, we compare against SOTA alignment-based method RLAIF-V and contrastive decoding methods VCD, M3ID, and AVISC, on the AMBER and MSCOCO datasets, as presented in Tab. 1, Tab. 2. Additionally, we evaluate our method against VCD on the LLaVA-QA90 dataset, presented in Tab. 3. Our method outperforms most existing approaches across various baseline models and datasets, highlighting its robustness and reliable performance across different data types and model architectures. Specifically, we surpass image-level contrastive decoding methods like VCD and M3ID, demonstrating its effectiveness in operating at the token level and adapting to individual samples. Furthermore, compared to the token-level AVISC, our method excels, likely due to its object awareness and adaptability to fluctuating bias levels. The results further demonstrate that our method effectively mitigates hallucinations while preserving information.

Table 1: Results on the AMBER Dataset.

| MLLMs | Method | Generative | | | | Discriminative | | | | Score ↑ |
|---|---|---|---|---|---|---|---|---|---|---|
| | | cha ↓ | cov ↑ | hal ↓ | cog ↓ | acc ↑ | P ↑ | R ↑ | F1 ↑ | |
| LLaVA v1.5 | base | 7.8 | **51.0** | 36.4 | 4.2 | 72.0 | **93.2** | 62.4 | 74.7 | 83.5 |
| | RLAIF-V 51 | 6.6 | 49.7 | 32.0 | 2.9 | 76.7 | 92.0 | 78.1 | 84.5 | 89.0 |
| | VCD 26 | 6.7 | 46.4 | 32.6 | 3.5 | 71.3 | 91.1 | 62.3 | 74.3 | 83.8 |
| | M3ID 12 | 6.2 | 50.5 | 29.3 | 2.8 | 72.4 | 91.8 | 64.1 | 75.5 | 84.7 |
| | AVISC 48 | 6.5 | 50.2 | 34.8 | 2.7 | 73.8 | 89.7 | 68.4 | 77.6 | 85.5 |
| | Ours | **5.6** | 51.0 | **24.3** | **1.8** | **80.3** | 82.9 | **89.3** | **86.0** | **90.2** |
| Instruct BLIP | base | 8.8 | **52.2** | 38.2 | 4.4 | 76.5 | 84.5 | 79.0 | 81.7 | 86.5 |
| | RLAIF-V 51 | 7.6 | 47.7 | 29.9 | 2.8 | 76.5 | 84.5 | 79.0 | 81.7 | 87.1 |
| | VCD 26 | 7.9 | 49.7 | 36.7 | 3.7 | 75.9 | 83.5 | 79.3 | 81.3 | 86.7 |
| | M3ID 12 | 7.3 | 49.2 | 33.8 | 3.7 | 75.8 | 84.4 | 77.9 | 81.0 | 86.9 |
| | AVISC 48 | 7.1 | 48.8 | 34.4 | 4.3 | 75.9 | 83.4 | **79.5** | 81.4 | 87.2 |
| | Ours | **6.0** | 49.4 | **26.6** | **2.4** | **78.1** | **88.8** | 76.6 | **82.2** | **88.1** |
| mPLUG Owl2 | base | 10.6 | 52.0 | 39.9 | 4.5 | 75.6 | **95.0** | 66.9 | 78.5 | 84.0 |
| | RLAIF-V 51 | 7.8 | 50.5 | 35.7 | 4.1 | 81.2 | 90.8 | 79.7 | 84.9 | 88.6 |
| | VCD 26 | 8.0 | 51.3 | 35.3 | 4.1 | 75.6 | 83.5 | 78.8 | 81.1 | 86.6 |
| | M3ID 12 | 7.8 | 51.7 | 34.9 | 4.1 | 75.9 | 83.5 | 79.3 | 81.3 | 86.8 |
| | AVISC 48 | 10.9 | 50.5 | 35.5 | 4.4 | 82.1 | 90.7 | 81.4 | 85.8 | 87.5 |
| | Ours | **7.5** | **53.6** | **34.7** | **4.0** | **82.1** | 87.0 | **86.2** | **86.6** | **89.6** |

**Hallucination Mitigation**. Our approach reduces hallucination by up to 33% at sentence-level ($hal$ in Tab. 1 and $C_S$ in Tab. 2) and 32% at instance-level ($cha$ in Tab. 1 and $C_I$ in Tab. 2), demonstrating its effectiveness in mitigating overall hallucinations. It also effectively mitigates co-occurrence hallucinations, with reductions of up to 57% for $cog$ in Tab. 1. Accuracy gains of up to 92% (Tab. 3)

---

[2]Since shorter outputs can trivially lower CHAIR scores at the expense of informativeness.

Table 2: Results Using the CHAIR Metric on the MSCOCO Subset Following Deng et al. (2024).

| Method | LLaVA-v1.5 | | | | InstructBLIP | | | | mPLUG-Owl2 | | | |
|---|---|---|---|---|---|---|---|---|---|---|---|---|
| | $C_S \downarrow$ | $C_I \downarrow$ | $R \uparrow$ | $Len \uparrow$ | $C_S \downarrow$ | $C_I \downarrow$ | $R \uparrow$ | $Len \uparrow$ | $C_S \downarrow$ | $C_I \downarrow$ | $R \uparrow$ | $Len \uparrow$ |
| base | 48.8 | 13.4 | **78.6** | 99.8 | 57.8 | 16.5 | **73.6** | 101.3 | 59.2 | 17.6 | **75.8** | 105.3 |
| VCD | 44.8 | 12.8 | 76.8 | 89.8 | 63.4 | 19.6 | 71.2 | 95.5 | 52.7 | 16.1 | 73.2 | 93.6 |
| M3ID | 44.5 | 12.1 | 77.0 | 85.1 | 57.3 | 16.1 | 72.5 | 100.1 | 52.4 | 15.8 | 72.7 | 92.6 |
| AVISC | 46.4 | 13.4 | 76.3 | 90.5 | 58.9 | 17.8 | 70.6 | 99.6 | 58.3 | 17.5 | 75.6 | 99.5 |
| Ours | **41.0** | **10.9** | 77.3 | 85.0 | **47.4** | **13.4** | 72.3 | 93.9 | **45.0** | **12.4** | 74.9 | 83.5 |

further demonstrate that our model improves alignment with the input image, highlighting its ability to jointly address text–visual and co-occurrence biases (Sec. 4.4).

**Information Preservation**. Our method also enhances information preservation, with recall (*cov* in Tab. 1 and $R$ in Tab. 2) dropping by an average of only $1.1\%$, compared to an average drop of $3.2\%$ in other methods, and even increasing by $3.1\%$ when using the baseline mPLUG-Owl2 on the AMBER dataset. Higher recall indicates that our model retrieves a broader range of objects from visual inputs. Additionally, results in Tab. 3 an increase of up to $45\%$ in detailedness (*Det*), further demonstrating our method's effectiveness in retrieving all relevant visual details and mitigating visual forgetting.

## 4.2 RESULTS ON DISCRIMINATIVE TASK

We next evaluate our method on discriminative tasks using AMBER (discriminative VQA) and POPE (existence VQA), with results shown in Tab. 1 and Tab. 4. Our approach achieves a better balance between precision and recall, yielding consistently higher F1 scores and improved accuracy. Notably, unlike competing methods that degrade Intern-VL2, ours preserves its performance via bias awareness, though improvements differ across categories and MLLMs.

Table 3: Results on LLaVA-QA90 Dataset, with All Settings Following Leng et al. (2024).

| Method | LLaVA-v1.5 | | InstructBLIP | | mPLUG-Owl2 | |
|---|---|---|---|---|---|---|
| | Acc↑ | Det↑ | Acc↑ | Det↑ | Acc↑ | Det↑ |
| base | 3.23 | 3.54 | 3.84 | 4.07 | 4.07 | 4.33 |
| VCD | 4.15 | 3.85 | 4.23 | 4.69 | 4.52 | 4.64 |
| Ours | **6.20** | **5.13** | **6.28** | **4.77** | **6.69** | **6.28** |

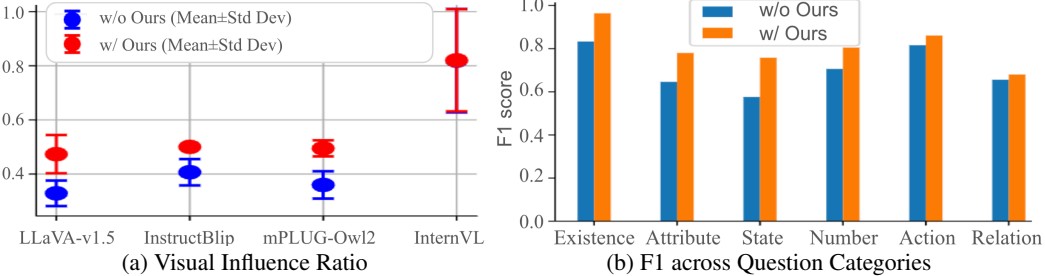

(a) Visual Influence Ratio    (b) F1 across Question Categories

Figure 3: (a) Visual influence ratios across the POPE dataset, illustrating variation across MLLMs. Our method successfully increases the visual influence ratio when it falls below $50\%$. (b) F1 scores for the AMBER discriminative task using LLaVA-v1.5 are consistently improved by our method, with particularly notable gains in the existence and state categories.

**Variation in Improvement Across Question Categories**. Fig. 3b presents F1 scores across various question categories using LLaVA-v1.5 Liu et al. (2024a). Our method improves performance across all categories, with the largest gains in existence, attributes,

Table 4: Results on POPE in MSCOCO Adversarial Setting.

| Method | LLaVA-v1.5 | | InstructBLIP | | mPLUG-Owl2 | | InternVL2 | |
|---|---|---|---|---|---|---|---|---|
| | Acc↑ | F1↑ | Acc↑ | F1↑ | Acc↑ | F1↑ | Acc↑ | F1↑ |
| base | 79.4 | 81.6 | 79.8 | 81.4 | 72.5 | 77.5 | **85.8** | **85.0** |
| VCD | 80.9 | 81.3 | 79.6 | 79.5 | 74.2 | 78.8 | 83.2 | 82.2 |
| M3ID | 81.7 | 81.8 | 81.0 | 81.6 | 75.6 | 79.1 | 83.5 | 82.1 |
| AVISC | 81.2 | 81.6 | 81.8 | 81.9 | 80.9 | 79.7 | 85.3 | 84.6 |
| Woodpecker | 80.5 | 80.6 | 79.0 | 78.6 | 77.5 | 76.9 | 85.7 | 84.8 |
| Ours | **83.5** | **82.1** | **82.5** | **82.1** | **84.2** | **83.7** | **85.8** | **85.0** |

and state—categories strongly tied to visual cues, benefiting from enhanced visual token influence.

**Variation in Improvement Across MLLMs**. Our method achieves the most significant improvement on mPLUG-Owl2 in object existence VQA (Tab. 4) and on LLaVA-v1.5 across various VQA categories (Tab. 1 discriminative), with minor improvement observed on InternVL2. This variation in performance correlates with the baseline visual influence ratios of the MLLMs. Fig. 3a presents the visual influence ratios in object existence VQA, showing that LLaVA-v1.5 exhibits the lowest visual contribution, followed by mPLUG-Owl2. This lower baseline visual influence ratio allows

378 our method to make more impactful adjustments. In contrast, InternVL2 has an original visual in-
379 fluence ratio exceeding $50\%$, resulting in minimal improvement when our method is applied. The
380 strong performance of InternVL2 can be attributed to its original high visual influence ratio, further
381 validating the motivation behind our approach. However, this dominant visual influence ratio in
382 InternVL2 is not consistent across other question categories or in open-ended generation tasks [3],
383 which allows our method to enhance its performance on the AMBER dataset (Tab. 6).

## 4.3 RESULTS ON ADDITIONAL MLLMS AND DATASETS

We further evaluate modern MLLMs, LLaVA-v1.6, Qwen2-VL and Intern-VL2, on the Amber
dataset. As shown in Tab. 6 (supplementary), our method consistently improves performance, even
for strong baselines. To test generality, we also evaluate on MMBench Liu et al. (2025), MMVet Yu
et al. (2023), ScienceQA Lu et al. (2022), and VizWiz Gurari et al. (2018). Results (supplementary)
show that our approach enhances perception and recognition, especially for tasks with clear visual
grounding, while preserving overall task performance and demonstrating robustness.

## 4.4 ABLATION STUDY

In this section, we first analyze text–visual and co-occurrence biases, and then study the contribu-
tions of our proposed components. Further details on gradient computation methods, norm selection,
and hyperparameter settings are provided in the supplementary material.

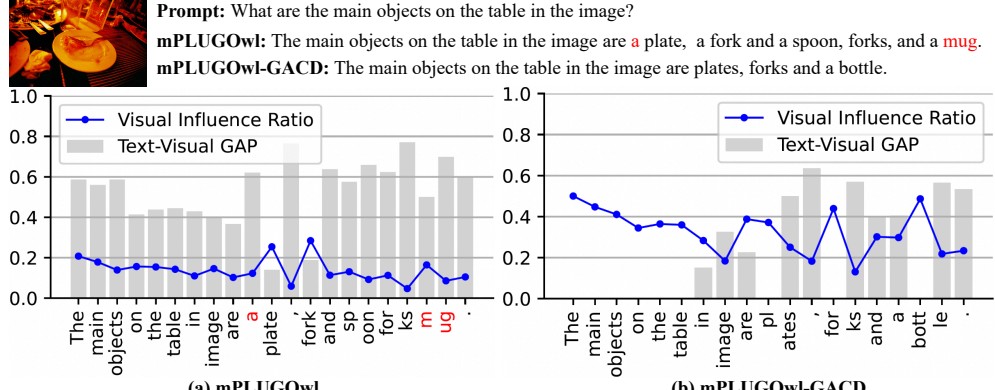

Figure 4: Comparison of visual influence ratios $r_m^v$ and Text-Visual GAP, with and without our
GACD. (a) Without GACD, mPLUG-Owl2 shows a low initial visual influence ratio, punctuation
marks and suffixes naturally have low visual influence, while objects start with higher influence
that declines as the sequence grows. Hallucinations tend to occur when the visual ratio is low. The
text-visual gap confirms that text dominates the influence on predictions. (b) With GACD, the visual
influence ratio increases overall and mitigates the decrease over the sequence length. The text tokens
only domain influence in predictions less related to the visual, reducing hallucination.

**Text-Visual Bias Analysis**. Fig. 3a shows that with the exception of InternVL2, MLLMs – LLaVA-
v1.5, InstructBLIP, and mPLUG-Owl2 – rely more on text prompt than on visual input. Likely
due to MLLMs' training process, this tendency is common in MLLMs, where multimodal fea-
tures are aligned with language tokens after extensive text-based pre-training, causing language
components to dominate decision-making. GACD effectively increases overall visual influence
to match that of object-present question prompts in POPE (Fig. 3a). In the open-ended genera-
tion task, we further observe the visual influence ratio $r_m^v$ and the Text-Visual GAP, defined as
$\max(\max(r_m^p, r_m^y) - r_m^v, 0)$, the difference between the text influence ratio and the visual influence
ratio when text influence is dominant [4]. Observations in Fig. 4 also highlight the text-dominant
influence typical of MLLMs. GACD counteracts this by boosting the influence of visual tokens
when aligning them with prompts and previous outputs, leading to higher prediction confidence and
a reduction in hallucinations (see Fig. 6 in supplementary). Additionally, the nature of the output
token affects the degree of visual influence ratio. For instance, punctuation marks or suffixes tend to

---

[3]Results for other question categories and visualizations of open-ended generation are in supplementary.

[4]$r_m^p$ and $r_m^y$ are derived in the same manner as $r_m^v$ in equation 14.

have a lower visual influence ratio. This is intuitive, as these tokens rely more on linguistic context and are less dependent on visual information. This observation highlights the value of our GACD framework delivering sample-dependent, token-specific hallucination mitigation.

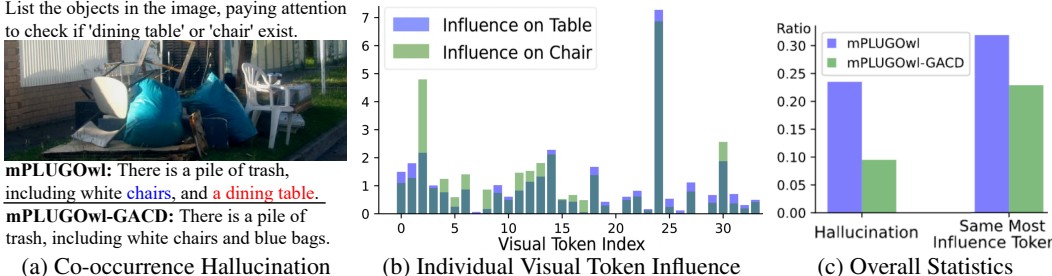

(a) Co-occurrence Hallucination     (b) Individual Visual Token Influence     (c) Overall Statistics

Figure 5: Co-occurrence hallucination of 'table' in the presence of 'chair'. (a) Comparison of outputs with and without GACD. (b) Visualization of individual visual token influence indicates that the visual token with index 24, which has the highest influence on the hallucinated 'table', also holds the highest influence on 'chair'. (c) Summary statistics for 100 chair-only and 100 table-only images, showing the hallucination rate and the percentage of cases where both objects share the same most influential visual token (as illustrated in b). GACD effectively reduces both metrics.

**Co-occurrence Bias Analysis**. Fig. 5a shows an example where mPLUG-Owl2 incorrectly predicts 'dining table' due to the presence of a 'chair'. In Fig. 5b, the influence of individual visual tokens on hallucinated prediction $I_{ms}$('table') and $I_{ms}$('chair') shows that they share the same most influential visual token: $s = 24$. We further collected 100 chair-only and 100 table-only images from MSCOCO evaluation dataset Lin et al. (2014). Results in Fig. 5c indicate that when only 'chair' or 'table' exists in the image, the hallucination rate for the other object is $23.5\%$, with a $31.9\%$ rate of sharing the same most influential token. This indicates that the 'Same Most Influential Token' phenomenon is common in co-occurrence hallucinations. Our GACD effectively reduces the hallucination where both 'table' or 'chair' are predicted in single-object images.

**Component Analysis**. To assess the effectiveness of each component in our proposed method, we conducted the following variants: 1) Visual Amplification (VA) only: visual amplification is applied to all visual tokens ($\mathbf{t}^v$), including during noun predictions. 2) Co-occurrence Hallucination Reduction (CR): object-related visual tokens are detected, and $\mathbf{t}^u$ is amplified during noun predictions. 3) Our full model, with early stopping (ES). Tab. 5 demonstrates that each component of our method contributes to the overall performance. VA significantly reduces hallucinations while improving object recall. CR further mitigates co-occurrence bias, a residual form of the text-visual bias addressed by VA, resulting in additional hallucination reduction. Both VA and CR achieve these gains without introducing trade-offs. When necessary, the ES mechanism shortens outputs to effectively reduce hallucinations, with only a slight recall trade-off that remains acceptable for most MLLMs.

Table 5: Component Analysis Using the CHAIR Metric.

| Components | | | LLaVA-v1.5 | | | | InstructBLIP | | | | mPLUG-Owl2 | | | |
|---|---|---|---|---|---|---|---|---|---|---|---|---|---|---|
| VA | CR | ES | $C_S \downarrow$ | $C_I \downarrow$ | $R \uparrow$ | $Len \uparrow$ | $C_S \downarrow$ | $C_I \downarrow$ | $R \uparrow$ | $Len \uparrow$ | $C_S \downarrow$ | $C_I \downarrow$ | $R \uparrow$ | $Len \uparrow$ |
| | | | 48.8 | 13.4 | 78.6 | 99.8 | 57.8 | 16.5 | 73.6 | 101.3 | 59.2 | 17.6 | 75.8 | 105.3 |
| ✓ | | | 46.4 | 11.6 | 79.0 | 95.6 | 53.6 | 15.1 | 75.3 | 108.4 | 52.6 | 14.4 | 78.2 | 95.6 |
| ✓ | ✓ | | 46.2 | 11.3 | 79.4 | 95.5 | 53.2 | 14.0 | 74.6 | 105.7 | 52.3 | 14.2 | 78.0 | 95.5 |
| ✓ | ✓ | ✓ | 41.0 | 10.9 | 77.3 | 85.0 | 47.4 | 13.4 | 72.3 | 93.9 | 45.0 | 12.4 | 74.9 | 83.5 |

## 5 CONCLUSION

In conclusion, we introduce a gradient-based self-reflection method to estimate token influence and quantitatively estimate bias severity. This estimation enables the identification of object-related visual tokens, which are then integrated into an influence-aware constrained decoding framework. This framework effectively mitigates both text-visual and co-occurrence biases, reducing hallucinations. Our method operates without requiring additional resources such as costly fine-tuning, extra models, or data statistics. Furthermore, to reduce text-visual bias in long-generated sequences, we propose a sample-dependent stopping criterion based on the proposed visual influence.

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

## A  FIRST ORDER TAYLOR EXPANSION

Let $\mathbf{z}_m^\star \in \mathbb{R}^{|\mathcal{V}|}$ denote the step-$m$ logits $\mathbf{z}_m^\star = \pi_{\theta^\star}(\mathbf{t}^v, \mathbf{t}^p, \mathbf{y}_{<m})$. Around a reference point $(\mathbf{t}^{v(0)}, \mathbf{t}^{p(0)}, \mathbf{y}_{<m}^{(0)})$, the detailed first-order Taylor expansion of the logit $\mathbf{z}_m^\star$ is

$$
\mathbf{z}_m^\star \approx \underbrace{\mathbf{z}_m^{\star(0)}}_{\pi_{\theta^\star}(\mathbf{t}^{v(0)}, \mathbf{t}^{p(0)}, \mathbf{y}_{<m}^{(0)})} + \sum_{s=1}^{S} \mathbf{g}_{ms}^v \left( \mathbf{t}_s^v - \mathbf{t}_s^{v(0)} \right)
$$

$$
+ \sum_{n=1}^{N} \mathbf{g}_{mn}^p \left( \mathbf{t}_n^p - \mathbf{t}_n^{p(0)} \right) + \sum_{i=1}^{m-1} \mathbf{g}_{mi}^y \left( \mathbf{y}_i - \mathbf{y}_i^{(0)} \right)
$$

$$
= \sum_{s=1}^{S} \mathbf{g}_{ms}^v t_s^v + \sum_{n=1}^{N} \mathbf{g}_{mn}^p t_n^p + \sum_{i=1}^{m-1} \mathbf{g}_{mi}^y y_i \tag{15}
$$

$$
+ \underbrace{\mathbf{z}_m^{\star(0)} - \sum_{s=1}^{S} \mathbf{g}_{ms}^v t_s^{v(0)} - \sum_{n=1}^{N} \mathbf{g}_{mn}^p t_n^{p(0)} - \sum_{i=1}^{m-1} \mathbf{g}_{mi}^y y_i^{(0)}}_{Const},
$$

$$
= \sum_{s=1}^{S} \mathbf{g}_{ms}^v t_s^v + \sum_{n=1}^{N} \mathbf{g}_{mn}^p t_n^p + \sum_{i=1}^{m-1} \mathbf{g}_{mi}^y y_i + Const,
$$

where the (token-wise) Jacobians are

$$
\mathbf{g}_{ms}^v \coloneqq \left. \frac{\partial \mathbf{z}_m^\star}{\partial \mathbf{t}_s^v} \right|_{\mathbf{t}^v = \mathbf{t}^{v(0)}}, \qquad \mathbf{g}_{mn}^p \coloneqq \left. \frac{\partial \mathbf{z}_m^\star}{\partial \mathbf{t}_n^p} \right|_{\mathbf{t}^p = \mathbf{t}^{p(0)}}, \qquad \mathbf{g}_{mi}^y \coloneqq \left. \frac{\partial \mathbf{z}_m^\star}{\partial \mathbf{y}_i} \right|_{\mathbf{y} = \mathbf{y}_{<m}^{(0)}}, \tag{16}
$$

and $\mathbf{z}_m^{\star(0)} = \pi_{\theta^\star}(\mathbf{t}^{v(0)}, \mathbf{t}^{p(0)}, \mathbf{y}_{<m}^{(0)})$. Here $\big|$ denotes evaluation at the reference point. Each $\mathbf{g}_{ms}^v, \mathbf{g}_{mn}^p, \mathbf{g}_{mi}^y$ maps a small token perturbation in its corresponding embedding space to a perturbation of the logit vector in $\mathbb{R}^{|\mathcal{V}|}$. And $Const$ denotes all other terms that are constant w.r.t., the $\mathbf{t}^v, \mathbf{t}^p$.

## B  INTERPRETING CONTRASTIVE DECODING THROUGH KL DIVERGENCE

Kullback-Leibler (KL) divergence can be used to interpret contrastive decoding, It measures the divergence between the reference distribution $p_{\theta^\star}(y_{cm}|\mathbf{t}^o, \mathbf{t}^p, y_{<m})$ to the $\mathbf{t}^u$ joint distribution $p_{\theta^\star}(y_{cm}|\mathbf{t}^v, \mathbf{t}^p, y_{<m})$, where $\mathbf{t}^v = \{\mathbf{t}^u, \mathbf{t}^o\}$.

$$
\mathrm{D}_{KL} = \sum_c p_{\theta^\star}(y_{cm}|\mathbf{t}^v, \mathbf{t}^p, y_{<m}) \log\left( \frac{p_{\theta^\star}(y_{cm}|\mathbf{t}^v, \mathbf{t}^p, y_{<m})}{p_{\theta^\star}(y_{cm}|\mathbf{t}^o, \mathbf{t}^p, y_{<m})} \right)
$$

$$
= \sum_c p_{\theta^\star}(y_{cm}|\mathbf{t}^v, \mathbf{t}^p, y_{<m})(\log(p_{\theta^\star}(y_{cm}|\mathbf{t}^v, \mathbf{t}^p, y_{<m})) - \log(p_{\theta^\star}(y_{cm}|\mathbf{t}^o, \mathbf{t}^p, y_{<m})))
$$

$$
= \sum_c p_{\theta^\star}(y_{cm}|\mathbf{t}^v, \mathbf{t}^p, y_{<m})([\pi_{\theta^\star}(\mathbf{t}^v, \mathbf{t}^p)_m]_c - \log(\sum \exp(\pi_{\theta^\star}(\mathbf{t}^v, \mathbf{t}^p)_m)) \tag{17}
$$

$$
- [\pi_{\theta^\star}(\mathbf{t}^o, \mathbf{t}^p)_m]_c + \log(\sum \exp(\pi_{\theta^\star}(\mathbf{t}^o, \mathbf{t}^p)_m)))
$$

$$
= \sum_c p_{\theta^\star}(y_{cm}|\mathbf{t}^v, \mathbf{t}^p, y_{<m})(\underbrace{[\pi_{\theta^\star}(\mathbf{t}^v, \mathbf{t}^p)_m - \pi_{\theta^\star}(\mathbf{t}^o, \mathbf{t}^p)_m]_c}_{\text{adjustment term}} + Const),
$$

where $p_{\theta^\star}(y_{cm}|\mathbf{t}^v, \mathbf{t}^p, y_{<m}) = \sigma(\pi_{\theta^\star}(\mathbf{t}^v, \mathbf{t}^p)_m)$, $p_{\theta^\star}(y_{cm}|\mathbf{t}^o, \mathbf{t}^p, y_{<m}) = \sigma(\pi_{\theta^\star}(\mathbf{t}^o, \mathbf{t}^p)_m)$ and $c$ represents a class in the predefined vocabulary. The adjustment term increases the KL divergence, thereby emphasizing the impact of visual tokens.

## C  RESULT ON MODERN MLLMS

We further evaluate recent MLLMs, LLaVA-v1.6, Qwen2-VL, and Intern-VL2, on the AMBER dataset. As shown in Tab. 6, our method consistently improves performance, even when the base

models already achieve strong results. Moreover, it surpasses SOTA comparison methods. This improvement stems from the model's self-reflection ability, which effectively identifies biases in baseline MLLMs and adapts the outputs accordingly.

Table 6: Latest MLLMs on the AMBER Dataset.

| MLLMs | Method | Generative | | | | Discriminative | | | | Score ↑ |
|-------|--------|------|------|------|------|------|------|------|------|---------|
| | | cha↓ | cov↑ | hal↓ | cog↓ | acc↑ | P↑ | R↑ | F1↑ | |
| LLaVA v1.6 | base | 9.9 | 56.7 | 47.4 | 4.3 | 80.3 | 82.9 | **89.3** | 86.0 | 88.5 |
| | RLAIF-V | 9.0 | 53.6 | 46.1 | 3.42 | 80.8 | 83.9 | 88.9 | 86.3 | 88.6 |
| | VCD | 9.5 | 52.7 | 46.3 | 3.78 | 79.9 | 83.1 | 87.6 | 85.4 | 88.0 |
| | M3ID | 9.2 | 50.1 | 45.3 | 3.3 | 80.4 | 83.2 | 88.8 | 85.9 | 88.4 |
| | AVISC | 9.2 | 50.7 | 47.5 | 3.2 | 80.6 | 83.5 | 88.2 | 85.8 | 88.3 |
| | Ours | **8.7** | **58.3** | **43.8** | **2.5** | **81.2** | **85.2** | 88.8 | **87.0** | **89.2** |
| Qwen2 VL | base | 6.4 | 70.4 | 54.8 | 5.9 | 82.9 | **91.6** | 82.2 | 86.6 | 90.1 |
| | RLAIF-V | 5.8 | 69.4 | 54.1 | 5.5 | 83.5 | 91.2 | 82.6 | 86.7 | 90.4 |
| | VCD | 6.5 | 69.1 | 53.7 | 5.3 | 82.7 | 90.9 | 82.3 | 86.4 | 90.0 |
| | M3ID | 6.3 | 68.8 | 53.5 | 5.1 | 83.0 | 91.0 | 82.8 | 86.7 | 90.2 |
| | AVISC | 6.3 | 69.0 | 53.9 | 5.0 | 82.8 | 91.1 | 82.5 | 86.6 | 90.1 |
| | Ours | **4.9** | **71.8** | **44.7** | **3.7** | **84.4** | 88.1 | **89.2** | **87.1** | **91.1** |
| Intern VL2 | base | 8.1 | 69.6 | 59.0 | 5.2 | 84.0 | 87.3 | **88.8** | 88.0 | 90.0 |
| | RLAIF-V | 8.0 | 68.4 | 59.3 | 4.9 | 84.2 | 87.7 | 88.5 | 88.1 | 90.1 |
| | VCD | 8.5 | 68.7 | 58.6 | 5.0 | 82.9 | 87.0 | 88.4 | 87.7 | 89.6 |
| | M3ID | 8.4 | 69.2 | 58.9 | 5.4 | 83.7 | 86.8 | 88.4 | 87.6 | 89.6 |
| | AVISC | 8.4 | 68.9 | 59.1 | 4.8 | 84.0 | 87.7 | 86.8 | 87.2 | 89.4 |
| | Ours | **7.9** | **69.8** | **57.8** | **3.7** | **84.7** | **88.2** | 88.8 | **88.5** | **90.3** |

# D  MLLMs ARCHITECTURES

Tab. 7 shows detailed information about the vision encoder and LLM components of the MLLM architectures used in our experiments.

Table 7: Details of the used MLLM architectures.

| MLLMs | Vision encoder | LLM |
|-------|----------------|-----|
| LLaVA-v1.5 (7B) | CLIP-L-336px | Vicuna-v1.5-7B |
| LLaVA-v1.5-13B | CLIP-L-336px | Vicuna-v1.5-13B |
| LLaVA-v1.6 | CLIP-L-336px | Vicuna-v1.5-7B |
| InstructBLIP (7B) | BLIP-2 | Vicuna-v1.1-7B |
| InstructBLIP-13B | BLIP-2 | Vicuna-v1.1-13B |
| mPLUG-Owl2 | CLIP-L | LLaMA-2-7B |
| InternVL2-4B | InternViT-300M-448px | Phi-3-mini-128k-instruct |

# E  RESULTS ON MMBENCH

We further evaluate our method on MMBench Liu et al. (2025). The results in Tab. 8 indicate that our method improves the overall performance and achieves consistent improvements across MLLMs on Coarse Perception (CP). This outcome aligns with the intended effect of our method, as its focus on increasing visual influence is directly linked to improving coarse perception capabilities. For other metrics, our method yields minor improvements due to the possible reason that certain abilities, such as Logical Reasoning (LR), rely more on the language component of MLLMs and cannot be enhanced solely by increasing visual influence.

# F  RESULTS ON MM-VET

The evaluation on MM-Vet Yu et al. (2023) in Tab. 9 shows that our method achieves consistent overall (Total) improvement, along with enhancements in recognition (Rec) and Optical Character

Table 8: Results on MMBench Dataset.

| MLLMs | Method | Overall | CP | FP-S | FP-C | AR | LR | RR |
|---|---|---|---|---|---|---|---|---|
| LLaVA-V1.5 | base | **62.3** | 68.5 | **69.6** | **57.7** | 73.1 | **29.9** | 54.7 |
| | ours | 61.8 | **73.2** | 62.6 | 53.0 | **73.3** | 27.8 | **57.8** |
| mPLUG-Owl2 | base | 63.5 | 68.1 | **69.1** | **55.8** | **78.4** | 37.0 | 57.0 |
| | ours | **65.0** | **72.6** | 66.6 | 53.0 | 76.0 | **41.6** | **63.0** |

Recognition (OCR), indicating its effectiveness in improving visual recognition. However, its performance varies across other metrics, including knowledge (Know), generalization (Gen), spatial awareness (Spat), and math (Math), suggesting that our method, which focuses on token influence balancing, may not effectively enhance the generalization ability of MLLMs.

Table 9: Results on MM-Vet dataset.

| MLLMs | Method | Rec | OCR | Know | Gen | Spat | Math | Total |
|---|---|---|---|---|---|---|---|---|
| LLaVA-V1.5 | base | 32.9 | 20.1 | **19.0** | 20.1 | **25.6** | 5.2 | 28.0 |
| | ours | **38.9** | **24.9** | 15.0 | 15.5 | 24.9 | **7.7** | **28.9** |
| InstructBlip | base | 32.4 | 14.6 | 16.5 | **18.2** | **18.6** | 7.7 | 26.2 |
| | ours | **40.5** | **18.0** | **18.7** | 17.4 | 14.9 | 3.8 | **26.6** |
| mPLUG-Owl2 | base | 36.1 | 19.4 | **29.8** | 19.4 | 23.9 | **7.7** | 27.3 |
| | ours | **45.0** | **26.4** | 27.9 | **25.9** | **24.8** | 3.8 | **33.9** |

## G    RESULTS ON SCIENCEQA AND VIZWIZ

We evaluate our method on two complementary multimodal benchmarks. ScienceQA Lu et al. (2022) integrates images, textual context, and curriculum knowledge, requiring models to perform structured multimodal reasoning. VizWiz Gurari et al. (2018), in contrast, consists of visual questions collected from blind users and features real-world challenges such as low-quality images, conversational queries, and unanswerable cases. These datasets jointly assess both reasoning under structured multimodal contexts and robustness in unconstrained real-world settings. As shown in Table 10, our approach consistently improves over the LLaVA-1.5 baseline. These gains demonstrate the effectiveness of our hallucination mitigation strategy in enhancing visual grounding across both knowledge-driven and real-world VQA tasks.

Table 10: Comparison of LLaVA-1.5 and our method on ScienceQA and VizWiz datasets.

| MLLMs | Method | ScienceQA(%) ↑ | VizWiz(%) ↑ |
|---|---|---|---|
| LLaVA-V1.5 | base | 66.2 | 48.7 |
| | Ours | **68.7** | **52.8** |

## H    REVISITING THE ACCURACY–INFORMATIVENESS TRADE-OFF

In the main paper, we report recall and output length alongside CHAIR scores, since our objective is to evaluate models under a balance of *accuracy* and *informativeness*. This choice is deliberate: our early stopping mechanism can be tuned to shorten responses, which naturally reduces CHAIR scores but at the expense of recall and content richness. Consequently, the trade-off introduced by early stopping is an explicit design choice, and it can be adjusted depending on the requirements of a specific application.

Direct comparison with SOTA methods that omit recall and generation length is therefore not entirely fair. Our analysis confirms that CHAIR scores can drop substantially when outputs are truncated, underscoring the importance of jointly reporting recall and length to present a complete view of performance. Without these complementary metrics, lower CHAIR values may simply reflect shorter, less informative responses rather than genuine improvements in visual grounding. To enable a fairer comparison with prior work, we adjust our early stopping threshold to 12%. Under this setting, our method achieves lower CHAIR scores while maintaining competitive recall, thereby

outperforming both approaches. This demonstrates that our framework not only mitigates hallucination effectively but also preserves informativeness. Moreover, the adjustable nature of the early stopping mechanism ensures that users can flexibly select the optimal balance between accuracy and informativeness for their specific use cases.

Table 11: Comparison with SOTA Methods with 12% Early Stopping Threshold.

| MLLM | Method | $C_s \downarrow$ | $C_i \downarrow$ | R $\uparrow$ | Len $\uparrow$ |
|---|---|---|---|---|---|
| | base | 48.8 | 13.4 | **78.6** | **99.8** |
| | PAI Liu et al. (2024c) | 24.8 | 6.9 | - | - |
| LLaVA-v1.5 | Middle Jiang et al. (2025) | 25.0 | 6.7 | - | - |
| | Ours_ES_12% | **23.5** | **6.5** | 55.0 | 54.1 |

# I  RESULTS ON MME

Our evaluation on MME Fu et al. (2024) dataset is presented in Tab. 12. Our method achieves better overall (Total) results and equal or improved performance in existence and counting, demonstrating its effectiveness in object recognition. However, it does not improve position accuracy and exhibits varying behavior on color. This diversity may stem from the inherent capabilities of MLLMs, which cannot be solely enhanced through token influence balancing.

Table 12: Result on MME Dataset.

| MLLMs | Method | Existence $\uparrow$ | Count $\uparrow$ | Position $\uparrow$ | Color $\uparrow$ | Total $\uparrow$ |
|---|---|---|---|---|---|---|
| LLaVA-v1.5 | base | 190.0 | 140.0 | 128.3 | 155.0 | 613.3 |
| | ours | 190.0 | **153.3** | 128.3 | **163.3** | **634.9** |
| IntructBLIP | base | 180.0 | 55.0 | 50.0 | 130.0 | 415.0 |
| | ours | **185.0** | 55.0 | 50.0 | 130.0 | **420.0** |
| mPLUG-Owl2 | base | 170.0 | 145.0 | 73.3 | **158.3** | 546.6 |
| | ours | 170.0 | **150.0** | 73.3 | 150.0 | **548.3** |

# J  OTHER RESULTS OF POPE

Table 13: More Results on POPE Li et al. (2023b).

| Dataset | Setting | Method | LLaVA-v1.5 | | InstructBLIP | | mPLUG-Owl2 | |
|---|---|---|---|---|---|---|---|---|
| | | | Acc $\uparrow$ | F1 $\uparrow$ | Acc $\uparrow$ | F1 $\uparrow$ | Acc $\uparrow$ | F1 $\uparrow$ |
| MSCOCO | Random | base | 86.5 | 84.8 | 87.1 | 85.7 | 86.0 | 84.4 |
| | | ours | **86.8** | **85.1** | **87.9** | **86.8** | **87.9** | **87.1** |
| | Popular | base | 85.5 | 83.8 | 84.2 | 83.6 | 84.6 | 83.2 |
| | | ours | **85.6** | **84.0** | **85.0** | **84.3** | **86.4** | **85.7** |
| A-OKVQA | Random | base | 88.0 | **87.6** | 88.5 | 88.5 | 86.5 | 85.7 |
| | | ours | **88.1** | 87.4 | **88.8** | **88.8** | **88.4** | **88.1** |
| | Popular | base | 85.5 | 85.1 | 81.9 | 83.1 | 82.4 | 82.2 |
| | | ours | 85.5 | 85.1 | **82.3** | **83.4** | **85.1** | **85.3** |
| | Adversarial | base | 79.1 | 79.9 | 74.8 | 77.9 | 74.7 | 76.9 |
| | | ours | **79.5** | **80.1** | **75.3** | **78.2** | **78.2** | **79.9** |
| GQA | Random | base | 88.9 | 88.2 | 87.2 | 87.1 | 85.2 | 84.0 |
| | | ours | 88.9 | 88.2 | 87.2 | **87.2** | **86.1** | **85.0** |
| | Popular | base | 84.1 | 84.1 | 78.6 | 80.4 | 78.7 | 78.5 |
| | | ours | **84.2** | 84.1 | **78.8** | 80.4 | **81.0** | **80.5** |
| | Adversarial | base | 80.8 | 81.3 | 75.9 | 78.4 | 76.4 | 76.8 |
| | | ours | **81.1** | **81.6** | **76.1** | **78.5** | **79.2** | **79.1** |

We report our experimental results on the POPE dataset, in addition to MSCOCO and adversarial settings, in Tab. 13. The results indicate that our method improves performance across all baseline MLLMs, with more significant gains observed in the adversarial setting. This discrepancy likely arises because adversarial scenarios require models to rely more heavily on visual inputs, aligning with our method's focus on enhancing visual influence. Conversely, for popular and random objects, textual data often provides sufficient statistical information, reducing the necessity for increased visual input reliance.

## K  DIFFERENT SAMPLING STRATEGIES

Tab. 14 presents an ablation study on sampling strategies (non-greedy vs. greedy). We follow the non-greedy sampling setting of VCD Leng et al. (2024), where both top-p and temperature are set to 1. As shown, our method consistently improves performance across both sampling strategies.

Table 14: Ablation Study on Sampling Strategies on POPE MSCOCO Adversarial Dataset.

| strategy | Method | LLaVA-v1.5 | | InstructBLIP | | mPLUG-Owl2 | |
|---|---|---|---|---|---|---|---|
| | | Acc | F1 | Acc | F1 | Acc | F1 |
| non-greedy | base | $79.0_{\pm0.51}$ | $81.1_{\pm0.53}$ | $71.6_{\pm0.49}$ | $74.7_{\pm0.46}$ | $71.5_{\pm0.30}$ | $76.6_{\pm0.28}$ |
| | ours | $\mathbf{82.3}_{\pm0.27}$ | $81.1_{\pm0.31}$ | $\mathbf{82.2}_{\pm0.29}$ | $\mathbf{81.8}_{\pm0.25}$ | $\mathbf{83.2}_{\pm0.27}$ | $\mathbf{82.9}_{\pm0.26}$ |
| greedy | base | 79.4 | 81.6 | 79.8 | 81.4 | 72.5 | 77.5 |
| | ours | **83.5** | **82.1** | **82.5** | **82.1** | **84.2** | **83.7** |

## L  DIFFERENT MODEL SIZE

We evaluate our method on different model sizes, 7B and 13B, for LLaVA-v1.5 and InstructBLIP, as shown in Tab. 15. The results indicate consistent improvements across various model sizes. In each model series, the smaller model gets a larger performance boost. With our method, we can achieve high accuracy and detection rates with a smaller 7B model, outperforming a 13B model at its original performance level.

Table 15: Ablation Study on Model Size on LLaVA-QA90 Dataset.

| Mothod | LLaVA-v1.5 | | | | InstructBLIP | | | |
|---|---|---|---|---|---|---|---|---|
| | 7B | | 13B | | 7B | | 13B | |
| | Acc | Det | Acc | Det | Acc | Det | Acc | Det |
| base | 3.23 | 3.54 | 4.78 | 4.2 | 3.84 | 4.07 | 5.67 | 4.88 |
| ours | **6.20** | **5.14** | **7.36** | **6.5** | **6.28** | **4.77** | **6.42** | **5.99** |

## M  GRADIENT COMPUTATION DETAILS AND EFFICIENCY ABLATION

Table 16: Ablation Study on Gradient Methods on the POPE MSCOCO Adversarial Dataset

| Methods | | Accuracy | F1 | Average Speed(ms) |
|---|---|---|---|---|
| MPLUG-Owl2 | IG Enguehard (2023) | 83.4 | 82.9 | 20335 |
| | direct | 84.2 | 83.7 | 385 |

Our method obtains gradients directly through PyTorch's 'torch.autograd.grad' on input tokens, eliminating the need for manual derivations and facilitating straightforward reproduction. For comparison, we evaluate Integrated Gradients (IG) Enguehard (2023); Lundstrom et al. (2022); Kapishnikov et al. (2021) using the SIG Enguehard (2023) implementation; Table 16 presents this ablation.

In the table, "IG" denotes the SIG-based results, while "direct" refers to our torch.autograd.grad approach. Both methods yield comparable accuracy and F1-score, but the direct-gradient variant is substantially more efficient.

## N    NORM SELECTION FOR TOKEN INFLUENCE

To evaluate the impact of norm selection on token influence, we study L1 (Manhattan), L2 (Euclidean), and $L_\infty$ (infinity) norms. The L1 norm highlights individual token contributions, while the L2 norm captures overall influence. The $L\infty$ norm focuses on the most dominant token or channel. Results in Tab. 17 show that the L1 norm achieves the best performance, aligning with our intuition that it effectively captures influence magnitude across tokens and channels. In contrast, the L2 norm, by emphasizing overall contribution, can obscure individual token effects, and the $L_\infty$ norm, though capturing the strongest signal, fails to account for the broader token/channel influence.

Table 17: Norm Strategies on POPE MSCOCO Adversarial Dataset.

| Norm | LLaVA-v1.5 | | InstructBLIP | | mPLUG-Owl2 | |
|---|---|---|---|---|---|---|
| | Acc | F1 | Acc | F1 | Acc | F1 |
| L1 | **83.5** | **82.1** | **82.5** | **82.1** | **84.2** | **83.7** |
| L2 | 83.2 | 81.9 | 79.5 | 79.6 | 83.2 | 82.9 |
| $L_\infty$ | 83.4 | 82.0 | 82.1 | 81.8 | 80.8 | 80.6 |

## O    HYPER PARAMETER STUDY

**Maximum $\alpha_m$.** To determine the optimal maximum amplification factor ($\alpha_m$) and understand its impact on the model performance, we conducted a search over values from 1 to 6 using the LLaVA-v1.5 model on the POPE dataset for discriminative tasks. For open-ended generation tasks on a subset of the MSCOCO dataset following Deng et al. (2024), we observed garbled text when $\alpha_m$ was set to 5; therefore, we limited our search to values from 1 to 4. As shown in Tab. 18 and Tab. 19, discriminative task on POPE is less sensitive to the value of $\alpha_m$. The performance on the generative task gets improved while $\alpha_m$ increases but later drops. Therefore, in our experiments, the optimal maximum amplification factor ($\alpha_m$) is set to 5 and 3 for discriminative and generation tasks, respectively.

Table 18: $\alpha_m$ Study For Discriminative Task On POPE Li et al. (2023b) in MSCOCO Adversarial Setting.

| Maximum $\alpha_m$ | 1 | | 2 | | 3 | | 4 | | 5 | | 6 | |
|---|---|---|---|---|---|---|---|---|---|---|---|---|
| | Acc $\uparrow$ | F1 $\uparrow$ | Acc $\uparrow$ | F1 $\uparrow$ | Acc $\uparrow$ | F1 $\uparrow$ | Acc $\uparrow$ | F1 $\uparrow$ | Acc $\uparrow$ | F1 $\uparrow$ | Acc $\uparrow$ | F1 $\uparrow$ |
| LLaVA-v1.5 | 83.4 | 82.0 | 83.4 | 82.0 | 83.4 | 82.0 | 83.4 | 82.0 | 83.5 | 82.1 | 83.4 | 82.1 |

Table 19: Maximum $\alpha_m$ Study For Generation Task on the MSCOCO Subset.

| Maximum $\alpha_m$ | 1 | | | | 2 | | | | 3 | | | | 4 | | | |
|---|---|---|---|---|---|---|---|---|---|---|---|---|---|---|---|---|
| | $C_S\downarrow$ | $C_I\downarrow$ | $R\uparrow$ | Len | $C_S\downarrow$ | $C_I\downarrow$ | $R\uparrow$ | Len | $C_S\downarrow$ | $C_I\downarrow$ | $R\uparrow$ | Len | $C_S\downarrow$ | $C_I\downarrow$ | $R\uparrow$ | Len |
| LLaVA-v1.5 | 44.0 | 11.8 | 76.2 | 86.1 | 41.4 | 11.1 | 77.4 | 84.8 | 41.0 | 10.9 | 77.3 | 85.0 | 41.4 | 10.9 | 77.3 | 84.9 |

**Early Stopping Threshold**. To set the early stopping threshold properly, we conducted a search on a subset of the MSCOCO dataset subset following Deng et al. (2024). Recognizing that the visual influence ratio varies across models, we first analyze the sample-wise visual influence to identify an appropriate range for this study. By searching over the corresponding range, we show results in Tab. 20. These results demonstrate that varying the ES threshold primarily mediates the trade-off between recall and hallucination rate. Our goal is to have balanced recall ($R$) and instance-level hallucination ($C_I$), leading us to select thresholds of 7% for LLaVA-v1.5 and LLaVA-v1.6, 25% for InstructBLIP, 2.5% for mPLUG-Owl2 and 10% for InternVL2. We additionally ran an experiment

to measure the ES activation rate using LLaVA-v1.5 with ES threshold $7\%$. As shown in Tab. 21, ES fires on only $8.7\%$ of the test samples, and when it does, the generated responses are on average just 0.7 tokens shorter. This indicates that ES rarely, and only minimally, truncates outputs.

Table 20: Early Stopping Threshold Study on the MSCOCO Subset

| | LLaVA-v1.5 | | | | LLaVA-v1.6 | | | | IntructBLIP | | | | mPLUG-Owl2 | | | | InternVL2 | | | |
|---|---|---|---|---|---|---|---|---|---|---|---|---|---|---|---|---|---|---|---|---|
| | 6% | 7% | 8% | 9% | 5% | 7% | 9% | 11% | 15% | 20% | 25% | 30% | 2% | 2.5% | 3% | 3.5% | 8% | 10% | 12% | 14% |
| $C_S$ | 45.6 | 41.0 | 36.6 | 31.4 | 29.0 | 26.0 | 23.0 | 17.8 | 52.6 | 51.4 | 47.4 | 36.0 | 51.8 | 45.0 | 41.2 | 41.0 | 37.0 | 35.2 | 34.6 | 32.9 |
| $C_I$ | 11.5 | 10.9 | 10.2 | 10.2 | 8.5 | 8.1 | 7.8 | 7.5 | 15.0 | 14.3 | 13.4 | 11.7 | 13.7 | 12.4 | 11.0 | 11.0 | 8.6 | 8.1 | 8.0 | 7.9 |
| $R$ | 79.7 | 77.3 | 75.2 | 70.8 | 68.5 | 63.0 | 58.8 | 53.0 | 75.1 | 74.4 | 72.3 | 68.8 | 77.7 | 74.9 | 73.8 | 73.5 | 65.8 | 65.4 | 65.4 | 64.0 |
| $L_{en}$ | 92.0 | 85.0 | 75.4 | 63.9 | 119.1 | 101.8 | 81.1 | 62.7 | 107.9 | 103.4 | 93.9 | 74.3 | 89.1 | 83.5 | 78.9 | 77.8 | 180.4 | 175.5 | 170.6 | 162.2 |

Table 21: Activation Rate of the Early Stopping on the MSCOCO Subset

| Methods | | Activate Percentage | Average Length |
|---|---|---|---|
| LLaVA-v1.5 (7%) | base | - | 85.1 |
| | ours | 8.7% | 84.4 |

## P    CONFIDENCE AND VISUAL INFLUENCE

Low confidence often signals potential failure modes in base MLLMs. Here, we demonstrate that our method not only improves accuracy but also increases model confidence. It remains effective even in low-confidence regions for three reasons: 1) We aggregate token gradients at the component level (Eq. 7) rather than using individual token gradients which yields robustness against local gradient noise. 2) We adjust influence towards visual tokens which consistently reduces the hallucination likelihood; 3) Empirically, low model confidence does not correlate with noisy gradients. In our experiments, pretrained MLLMs usually maintain meaningful gradient signals even at low confidence levels. Fig. 6 shows an example where the baseline model mPLUG-Owl2 exhibits low confidence in hallucinated predictions and near-zero confidence in the initial predictions for 'forks' and 'mug'. With GACD, prediction confidence increases alongside the visual influence ratio, with the minimum confidence rising to over $30\%$.

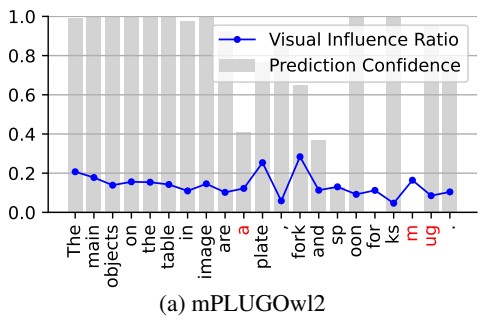
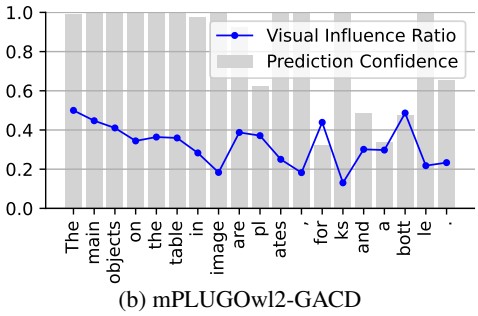

(a) mPLUGOwl2                                  (b) mPLUGOwl2-GACD

Figure 6: Comparison of prediction confidence with and without GACD. (a) Without GACD, mPLUGOwl2 exhibits low confidence in hallucinated predictions and near-zero confidence in the initial predictions for 'forks' and 'mug'. (b) With GACD, mPLUGOwl2's confidence increases alongside the visual influence ratio, effectively mitigating hallucinations.

## Q    QUESTION CATEGORY RESULTS ON THE AMBER DATASET

We report discriminative results across different question categories in Tab. 22. Our method improves performance in nearly all categories across all MLLMs. The improvement in InternVL2's object existence is minor, likely due to its already high visual influence ratio. For LLaVA-v1.5

and mPLUG-Owl2, which have lower original visual influence ratios, our method achieves more substantial gains in existence, attribute, and state categories.

Table 22: Results on the Question Categories of Discriminative Task on AMBER Dataset.

| Category | Metric | InstructBLIP | | LLaVA-v1.5 | | LLaVA-v1.6 | | mPLUG-Owl2 | | Intern-VL2 | |
| | | base | ours | base | ours | base | ours | base | ours | base | ours |
|---|---|---|---|---|---|---|---|---|---|---|---|
| Existence | acc | 70.0 | 79.8 | 70.8 | 93.2 | 92.9 | 93.0 | 75.2 | 89.9 | 90.6 | 90.6 |
| | P | 100.0 | 100.0 | 100.0 | 100.0 | 100.0 | 100.0 | 100.0 | 100.0 | 100.0 | 100.0 |
| | R | 70.0 | 79.8 | 70.8 | 93.2 | 92.9 | 93.0 | 75.2 | 89.9 | 90.6 | 90.6 |
| | F1 | 82.3 | 88.7 | 82.9 | 96.4 | 96.3 | 96.3 | 85.8 | 94.6 | 95.0 | 95.0 |
| Attribute | acc | 71.9 | 78.3 | 72.3 | 76.1 | 75.2 | 77.1 | 73.9 | 78.2 | 82.3 | 82.6 |
| | P | 76.0 | 81.7 | 87.3 | 74.0 | 74.6 | 76.4 | 86.0 | 76.9 | 80.9 | 80.9 |
| | R | 64.3 | 73.0 | 52.2 | 82.7 | 83.0 | 83.9 | 57.1 | 81.8 | 84.7 | 85.2 |
| | F1 | 69.7 | 77.1 | 65.3 | 78.1 | 78.5 | 80.0 | 68.6 | 79.3 | 82.8 | 83.0 |
| State | acc | 73.4 | 76.4 | 68.2 | 73.3 | 78.6 | 75.2 | 70.5 | 77.9 | 81.2 | 81.2 |
| | P | 75.1 | 77.1 | 86.2 | 70.3 | 78.6 | 74.7 | 84.9 | 75.5 | 79.1 | 78.7 |
| | R | 70.6 | 75.3 | 43.3 | 82.0 | 78.5 | 82.9 | 49.8 | 83.1 | 84.8 | 85.5 |
| | F1 | 72.8 | 76.2 | 57.6 | 75.7 | 78.5 | 78.6 | 62.8 | 79.1 | 81.8 | 82.0 |
| Number | acc | 65.4 | 80.6 | 75.0 | 80.1 | 80.1 | 80.2 | 77.8 | 76.5 | 82.6 | 83.3 |
| | P | 75.4 | 93.1 | 86.9 | 79.1 | 79.2 | 78.6 | 86.0 | 77.0 | 83.0 | 84.0 |
| | R | 45.8 | 66.2 | 59.5 | 82.4 | 81.7 | 84.4 | 66.9 | 77.0 | 82.0 | 82.3 |
| | F1 | 57.0 | 77.4 | 70.6 | 80.7 | 80.4 | 81.4 | 75.3 | 77.0 | 82.5 | 83.1 |
| Action | acc | 79.7 | 83.7 | 83.6 | 82.3 | 81.9 | 80.4 | 84.0 | 84.1 | 88.4 | 88.6 |
| | P | 82.5 | 88.5 | 92.9 | 85.9 | 79.4 | 81.2 | 90.9 | 85.9 | 86.5 | 86.8 |
| | R | 75.3 | 77.5 | 72.7 | 87.4 | 86.0 | 88.6 | 75.5 | 85.9 | 90.9 | 91.2 |
| | F1 | 78.7 | 82.6 | 81.6 | 86.6 | 82.6 | 84.7 | 82.5 | 85.9 | 88.6 | 88.9 |
| Relation | acc | 62.7 | 71.9 | 71.8 | 61.5 | 64.5 | 65.7 | 70.5 | 76.9 | 72.1 | 77.0 |
| | P | 56.2 | 64.0 | 65.9 | 51.9 | 54.0 | 56.6 | 61.0 | 67.9 | 60.0 | 65.1 |
| | R | 48.6 | 73.4 | 66.3 | 97.7 | 95.1 | 87.2 | 79.5 | 83.9 | 98.3 | 95.6 |
| | F1 | 52.1 | 68.4 | 66.1 | 67.8 | 68.9 | 68.6 | 69.0 | 75.1 | 74.5 | 77.5 |

## R  COMPUTATIONAL COST

In this section, we analyze the computational cost of the proposed method. On the POPE dataset, our method increases the average computational cost by $101.44\%$ in TFLOPs. This is expected for decoding-based approaches, as they require two forward passes. Importantly, the visual encoder is executed only once, and the second pass processes a limited number of input tokens. Thus, although additional computation is introduced, the overall overhead remains within a practical range.

Table 23: TFLOPs of Baseline vs. Ours with LLaVA-v1.5 on the POPE Dataset.

| Method | TFLOPs | Relative Increase |
|---|---|---|
| LLaVA-v1.5 | 9.68 | – |
| Ours | 19.49 | $+101.44\%$ |

## S  INFLUENCE RATIO IN VQA

Fig. 7 illustrates the visual influence ratio across outputs in VQA tasks, comparing baseline predictions with those obtained after applying GACD. The results confirm that text tokens dominate influence across MLLMs, including InternVL2, which exhibits a relatively high visual influence ratio. As shown in Fig. 3 of the main paper, the overall $60\%$–$100\%$ visual influence ratio across the POPE dataset suggests that visual inputs predominantly determine object existence in VQA tasks. GACD enhances visual influence, effectively balancing text-visual bias. Furthermore, the visualization on InternVL2 demonstrates that the co-occurrence hallucination 'knife' persists despite a high visual influence. GACD successfully eliminates this co-occurrence hallucination.

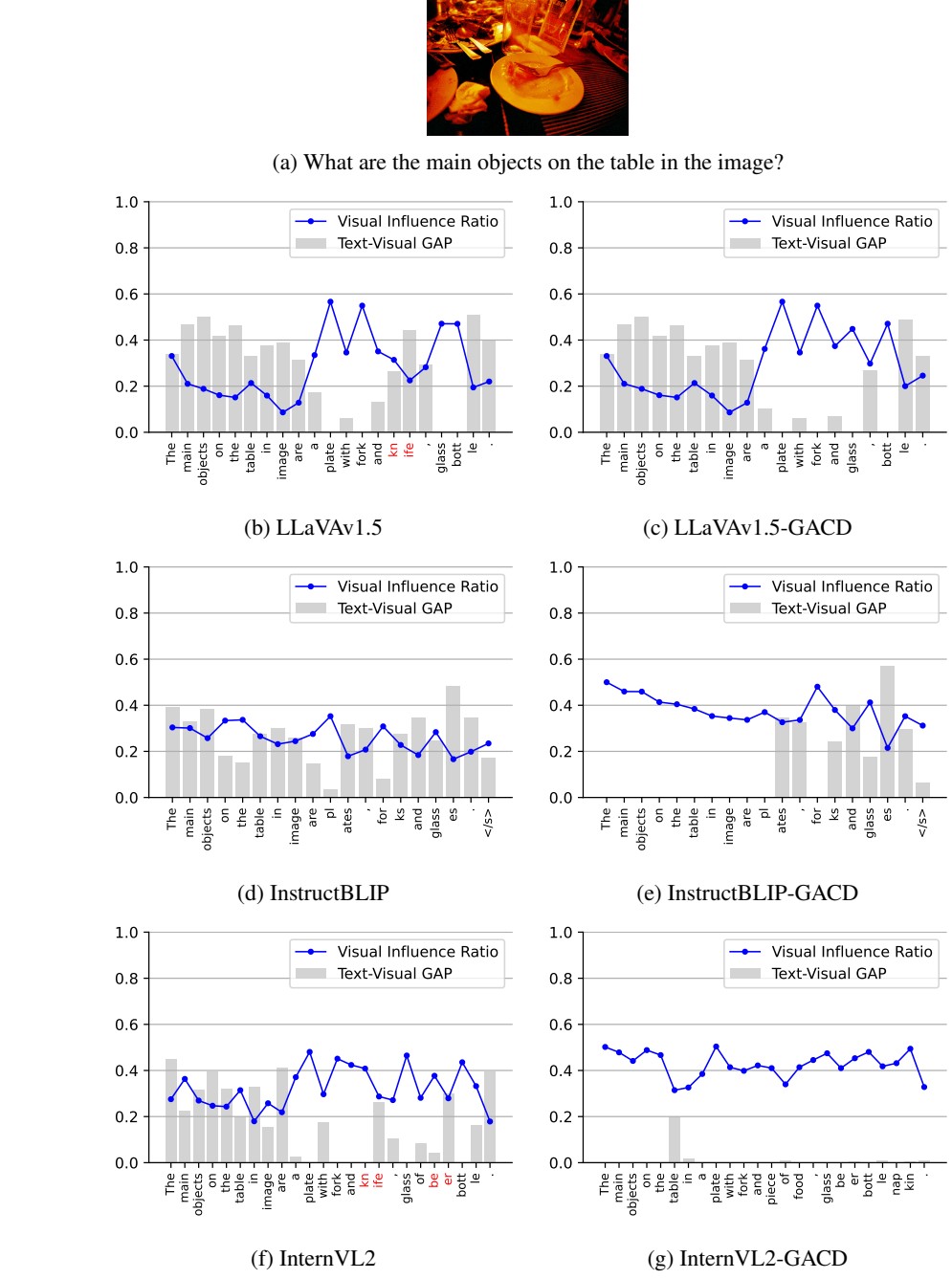

Figure 7: Influence Ratio across Predicted Tokens in VQA: (left) Baseline predictions; (right) Predictions with GACD. GACD effectively mitigate Text-Visual GAP, balancing text-visual bias. (f) The original InternVL2 shows a dominant visual influence ratio at the hallucinated prediction 'knife', indicating a co-occurrence bias that remains unaddressed even with dominant visual influence. (g) GACD successfully eliminates co-occurrence hallucinations, including 'knife'.

# T  INFLUENCE RATIO IN IMAGE CAPTION

We further visualize the influence ratio in the image captioning task. Fig. 8 shows that in the baseline LLaVA-v1.5, the influence gap between previous output tokens and visual tokens widens as more

tokens are generated. However, GACD effectively narrows this gap, preventing visual information from being forgotten and thereby reducing hallucinations.

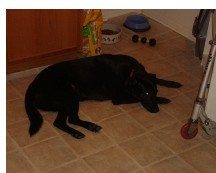

**LLaVA:** The image features a black dog lying on the floor in a kitchen, resting comfortably on a tiled surface. The dog is positioned near a bowl, possibly enjoying a meal or a snack. In the kitchen, there are several items scattered around. A bottle can be seen on the left side of the dog, while a cup is located closer to the center. A spoon is placed near the right side of the dog, and a bowl is situated on the right side of the scene.

**LLaVA-VA:** The image features a black dog lying on the floor of a kitchen, resting comfortably on a tile floor. The dog is positioned near a bowl, a toy, and a bag of cat food. The bowl is placed on the floor, while the toy is located closer to the dog.

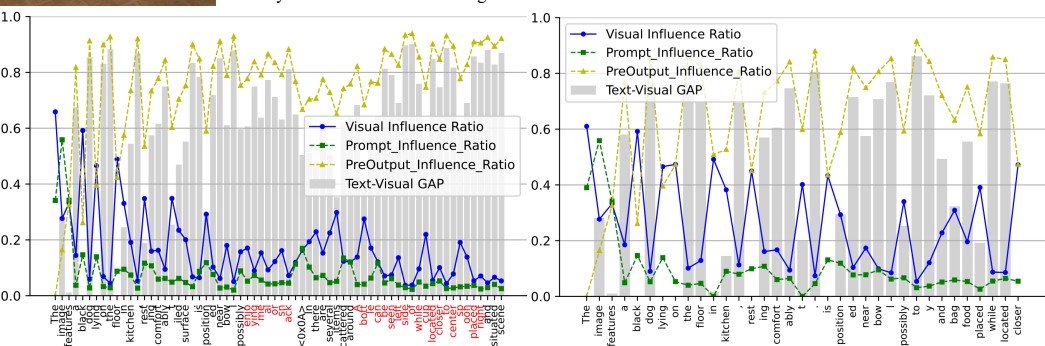

Figure 8: Comparison of influence ratios across predicted tokens with and without GACD. (Left) Without GACD, the influence gap between previous output tokens and visual tokens widens as more tokens are generated. (Right) With GACD, the gap is periodically narrowed to nearly zero, mitigating this trend and reducing hallucination.

## U    QUALITATIVE EXAMPLE ON OCCLUDED IMAGES

We include a qualitative example in Fig. 9, where a sedan and a building are partially occluded by a white truck, our method prevents the baseline model from hallucinating of persons and vehicles behind the occluding object. This demonstrates our method remains effective on images consisting of occlusions. Image caption w/o Ours: The image features a white truck parked on a city street,

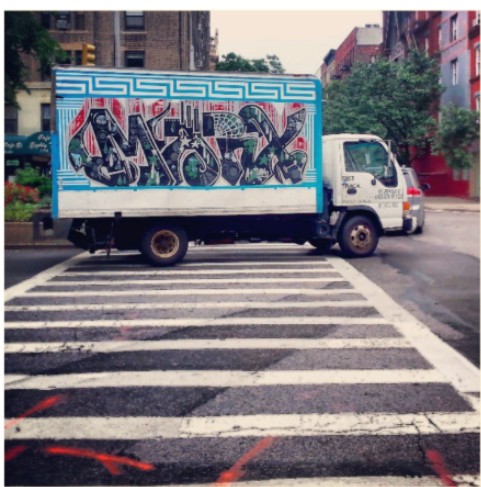

**Image caption w/o Ours:** The image features a white truck parked on a city street, with graffiti covering its side. The truck is positioned near a crosswalk, and there are several other vehicles in the scene, including a car and a bus. In addition to the vehicles, there are a few people walking around the area.

**Image caption w Ours:** The image features a large white truck parked on a city street, with graffiti covering its side. The truck is positioned near a crosswalk, and there is another vehicle visible in the background. Additionally, there are buildings in the scene, suggesting an urban setting.

Figure 9: Example of our method applied to an occluded image.

with graffiti covering its side. The truck is positioned near a crosswalk, and there are several other vehicles in the scene, including a car and a bus. In addition to the vehicles, there are a few people walking around the area. Image caption w Ours: The image features a large white truck parked on a city street, with graffiti covering its side. The truck is positioned near a crosswalk, and there is another vehicle visible in the background. Additionally, there are buildings in the scene, suggesting an urban setting.

## V    ADDITIONAL IMPLEMENTATION AND EXPERIMENTAL DETAILS

Noun tokens are identified using the spaCy library via its `en_core_web_sm` model. For experiments on the Amber dataset Wang et al. (2023), we adopt the original data splits and evaluation metrics. In the MSCOCO Lin et al. (2014) subset, we follow the data partitioning and evaluation protocol of Deng et al. Deng et al. (2024), with splits available in their official repository. For the LLaVA-QA90 Liu et al. (2024b), MME Fu et al. (2024) and POPE Li et al. (2023b) datasets, our setup replicates that of Leng et al. Leng et al. (2024) and use their provided scoring scripts for LLaVA-QA90. Experiments on MMBench Liu et al. (2025), MM-Vet Yu et al. (2023) follow the VLMEvalKit_InternVL2_5 repository. All comparison methods are executed using their official code; we only modify them to enforce greedy sampling and a uniform maximum generation length to align with our experimental settings.

## W    LIMITATIONS AND FUTURE WORK

Our method is limited to white-box MLLMs, as it requires access to gradients. Its effectiveness depends on the baseline MLLM's original visual influence ratio, and the importance of visual information. As a post-processing technique, our method does not involve model training. In future work, we aim to explore how insights from GACD can guide and improve training strategies for enhanced visual perception in MLLMs.

## X    BROADER IMPACTS

Our method enhances the factual reliability of multi-modal language models, not only for vision–language tasks but also for modalities such as video and audio, by mitigating hallucinations at inference time. This improvement has several positive societal implications: it can make systems for visual question answering, assistive technologies for the visually impaired, and automated image captioning more dependable, thereby increasing user trust and safety; it can power educational tools that generate accurate descriptions of complex diagrams or historical media, benefiting learners and instructors; and in critical domains such as medical imaging or remote sensing, it can reduce spurious outputs and support more robust decision-making. Conversely, if deployed within surveillance or facial-recognition systems, stronger multi-modal grounding could facilitate more intrusive inferences about individuals from visual data, exacerbating privacy risks.

