# OpenReview forum: "Mitigating Multimodal Hallucinations via Gradient-based Self-Reflection"
_ICLR.cc/2026/Conference — ICLR 2026 Conference Withdrawn Submission_

### Official Review · Reviewer_cdiL · 2025-10-27

**Soundness:** 3
**Presentation:** 3
**Contribution:** 3
**Rating:** 6
**Confidence:** 5

**Summary:**

The paper introduces Gradient-based Influence-Aware Constrained Decoding (GACD), an innovative inference-based method for mitigating hallucinations in multimodal large language models (MLLMs). It addresses two core biases causing hallucinations: text-visual bias (overreliance on text) and co-occurrence bias (spurious correlations in training data). The method calculates the influence of each token (both visual and textual) using first-order Taylor gradients, then modifies the token influences during inference to reduce hallucinations. It employs a two-part approach: (1) suppressing spurious visual features and (2) rebalancing the visual and text contributions to the generation.

**Strengths:**

1. The paper clearly identifies a significant problem, hallucinations in multimodal large language models, and provides a well-defined solution based on gradient-based token influence estimation.
2. The approach is computationally efficient and does not require additional models or fine-tuning. It can be directly applied to existing MLLMs, making it highly practical for improving real-time applications.
3. Extensive experiments demonstrate that GACD effectively reduces hallucinations and improves grounding in vision-based tasks (e.g., VQA, image captioning). The results on datasets like LLaVA-QA90 and AMBER confirm the utility of the approach across different MLLMs and task categories.

**Weaknesses:**

1. The method relies on calculating gradients to estimate token influence, but there is insufficient detail on the exact mechanics of how gradients are computed, particularly with respect to visual tokens. The alignment of visual and text tokens across different models is also not fully addressed.
2. The evaluation is limited to only a few baselines, and several important related methods (e.g., OPERA [1], MemVR [2]) are not included for direct comparison. Additionally, there is little discussion about how these methods might perform under matched conditions with GACD.
3. While the paper mentions the computational cost of the method, the results suggest a substantial increase in processing time.


[1] OPERA: Alleviating Hallucination in Multi-Modal Large Language Models via Over-Trust Penalty and Retrospection-Allocation, CVPR 2024.

[2] Look Twice Before You Answer: Memory-Space Visual Retracing for Hallucination Mitigation in Multimodal Large Language Models, ICML 2025.

**Questions:**

1. How would GACD scale when applied to much larger models (e.g., 13B or 13B+ parameters) or datasets requiring real-time inference? Is there a way to optimize the computational overhead further?
2. How do you determine the optimal threshold for early stopping? Is there a trade-off between visual grounding and informativeness, and how does the model balance them in different tasks?
3. Do you apply any gradient smoothing, moving average, or regularization to stabilize influence computation during decoding?
4. How sensitive is model performance to λ? Was any hyperparameter tuning or adaptive thresholding strategy considered?
5. Is β dynamically adjusted per decoding step or fixed globally? If dynamic, what metric determines the adjustment, e.g., attention entropy, gradient magnitude?

---

### Official Review · Reviewer_zLQM · 2025-10-31

**Soundness:** 1
**Presentation:** 2
**Contribution:** 2
**Rating:** 2
**Confidence:** 5

**Summary:**

Hallucinations in multimodal LLMs mainly arise from text–visual and co-occurrence biases. To address this, the paper introduces a gradient-based token influence estimation that quantifies how visual inputs, text prompts, and previously generated tokens affect the output. Based on these influence values, the proposed inference-based method suppresses object-related text bias while amplifying the overall visual token influence to enhance visual grounding.

**Strengths:**

- The introduction is well written and highly readable.
- The motivation behind the proposed method is clearly presented and easy to follow.
- The paper conducts extensive experiments across various backbones and benchmarks, along with diverse analyses.

**Weaknesses:**

1. **Limitation of the gradient-based inference method**
   * Although the proposed approach is described as inference-based, it still requires gradient computation. If gradient information is already accessible, it is unclear why a training-free approach is necessary in the first place.
2. **Weak linkage between theory and application (Section 3.2)**
   * The theoretical foundation and its practical utilization are not clearly connected. It would be helpful to either strengthen the explanation in Section 3.2 or integrate this part into the background section for a more coherent flow.
3. **Ambiguity in the Object-aware Visual Token Grouping section**
   * Since masking and grouping are applied only during “noun prediction”, how is this step distinguished at inference time? The procedure for detecting nouns in y is not explained. Does it require additional annotation or external tools?
   * The dimensions and aggregation process of the masks (\\mathcal{M}\_{is}​ and \\mathcal{M}\_{ms}) are unclear. The current description requires multiple readings to follow.
   * The paper excludes nouns in the question prompt, but such nouns can also carry strong object information that may induce co-occurrence bias. The rationale behind this design choice should be clarified.
4. **Potential side effects of weakening object-related visual tokens**
   * Objects are critical cues for image understanding. Suppressing object-related visual tokens might unintentionally harm visual grounding performance. A discussion or analysis on potential side effects would strengthen the work.
5. **Unclear interpretation of the Visual Influence Ratio**
   * It is not entirely convincing how a higher visual influence ratio in InternVL supports the claim of limited performance improvement. The main contribution lies not only in enhancing visual influence but also in object-aware token grouping.
   * Moreover, in Table 5, the combination of VA+CR shows lower performance than VA alone across all backbones and metrics. This makes it difficult to justify CR as a meaningful component. Please provide a clearer explanation of this result, and consider conducting additional experiments — for example, reporting results for VA+ES — to help isolate and verify the effect of each component.

**Questions:**

1. **On the computational cost in Appendix R**
   * Appendix R discusses the computational cost, but it would be helpful to include comparisons with other inference-based methods. The notion of a “practical range” feels rather subjective; showing relative cost differences with related techniques mentioned in prior work would make the discussion more convincing.
2. **Formatting and minor issues**
   * The paper appears to follow last year’s format.
   * Line 69: Typographical error.
   * Figure 2 caption is broken (“textcolordarkgreen”).
   * Line 218: Replace the comma with a period.
3. **Ambiguity in terminology and expressions**
   * The paper contains several ambiguous expressions that make it difficult to follow. For instance, in lines 246–248, t^u only appears in the original logit, while “other inputs” are said to contribute to z^o. Does “other inputs” refer specifically to visual inputs excluding the text inputs?
   * Moreover, the term “group influence” in line 248 is not clearly defined.
   * Clarifying these terms and their roles would improve the overall readability and conceptual clarity of the paper.

---

### Official Review · Reviewer_CECC · 2025-11-01

**Soundness:** 2
**Presentation:** 2
**Contribution:** 2
**Rating:** 2
**Confidence:** 4

**Summary:**

This paper challenges the view that attention sinks in MLLMs are merely errors, arguing instead that specific tokens within the system prompt act as functional "guardians" essential for model stability and reasoning. Through a series of causal interventions, the authors demonstrate that these sink tokens are crucial for computational stability and act as part of an internal "state machine" for complex tasks. Building on this insight, the paper proposes the "Attention-Budget Hypothesis" and introduces SPEAR, a hallucination mitigation method that preserves these critical sink tokens while reallocating attention from non-essential text to visual tokens. Experimental results show SPEAR effectively mitigates hallucination without compromising reasoning abilities, thus validating the functional importance of sink tokens.

**Strengths:**

1. The paper proposes a novel and principled approach to mitigate hallucinations by conceptualizing the problem through the lens of "gradient-based self-reflection." The idea of using first-order Taylor expansion to quantify the influence of individual visual and text tokens at inference time is elegant and provides a fine-grained, interpretable way to diagnose and address model biases without requiring fine-tuning. This influence-aware framework is insightful and could inspire future work on model interpretability and controllable generation.

2. The paper provides a clear and insightful analysis of the sources of multimodal hallucinations, effectively categorizing them into text-visual bias and co-occurrence bias. The proposed method, GACD, is well-designed to tackle both biases simultaneously. Furthermore, the experimental evaluation is comprehensive, covering multiple models, datasets, and tasks. The authors' explicit consideration of the trade-off between hallucination reduction and information preservation is a commendable aspect, as this critical balance is often overlooked in related work.

**Weaknesses:**

1. Upon examining Table-6 in the appendix, I noticed a significant discrepancy in the reported performance of the LLaVA-v1.5 baseline on the POPE benchmark. The results presented in this paper are considerably lower than the officially reported figures—for instance, on the MSCOCO Random and Popular splits, the accuracy is approximately 2 percentage points lower on average. This is concerning because the official LLaVA-v1.5 results are readily reproducible using the publicly available code. This performance gap raises serious questions about the fairness of the experimental comparison and suggests that the baseline may have been weakened, which in turn would exaggerate the relative improvements of the proposed method.

2. The core mechanism of GACD requires calculating gradients of the output logits with respect to all input visual and text tokens at every single decoding step. This involves a backward pass through a significant portion of the model for each generated token, which is computationally expensive and drastically increases inference latency. The paper fails to provide a thorough analysis of this overhead. For a method positioned as a practical, fine-tuning-free solution, this performance degradation is a major drawback that severely limits its applicability in real-time or resource-constrained scenarios. A critical comparison of inference speed (e.g., tokens per second) against standard decoding methods is missing.

3. The method's foundation rests on a first-order Taylor expansion to approximate the influence of a token: $ \Delta L \approx \nabla_{\mathbf{e}} L \cdot \Delta \mathbf{e}$. This linear approximation is only accurate for infinitesimally small changes around the current embedding point. However, the influence of a token on the final output in a deep, highly non-linear model like an MLLM is far more complex. The gradient only captures the local sensitivity and may not be a reliable proxy for the token's true, global contribution to the final semantics. The paper does not provide sufficient theoretical or empirical evidence to justify why this local, linear approximation is a robust measure for complex phenomena like co-occurrence bias across the entire generation process.

4. The method fundamentally assumes that high-gradient visual tokens are "grounding" and should be amplified, while high-gradient text tokens from the prompt might represent "bias" and should be suppressed. This is a strong and potentially flawed assumption. A high gradient on a visual token could equally stem from a spurious co-occurrence (e.g., the model is highly sensitive to "grass" when generating "cow," even if the cow is in a desert). Conversely, a high-gradient text token from the prompt might be a crucial instruction that should not be suppressed. The gradient magnitude itself is agnostic to whether the influence is factually correct or biased. By simply re-weighting based on magnitude, GACD risks incorrectly amplifying spurious visual features or suppressing critical textual context, potentially leading to a different kind of error.

5. While GACD is "fine-tuning-free," it is not "hyperparameter-free." The method introduces several critical hyperparameters, such as the thresholds for identifying "influential" tokens and the scaling factors used to amplify or suppress their contributions. The optimal values for these hyperparameters are likely to be highly dependent on the specific model architecture, the dataset, and the nature of the task. This requires a separate, labeled validation set for tuning, which undermines the "plug-and-play" appeal of the method. The paper lacks a thorough sensitivity analysis showing how performance varies with these hyperparameters, making the method's robustness and generalizability questionable.

**Questions:**

1. The experimental results presented in Table-6 of the appendix require significant clarification. As they stand, these results are unconvincing and fail to provide compelling evidence for the proposed method's effectiveness. The authors must provide a detailed explanation for these outcomes; otherwise, it is difficult for a reader to be persuaded of the method's claimed benefits.

2. Your paper positions GACD as a practical, fine-tuning-free solution. However, the core mechanism requires a backward pass for every generated token, which seems to introduce a significant computational overhead at inference time. Could you provide a quantitative analysis of this latency, for instance, by reporting the inference speed (e.g., tokens per second) and comparing it against standard decoding methods? Without this analysis, it is difficult to assess the method's practicality for real-world applications, especially in resource-constrained or real-time scenarios.

3. The method's theoretical foundation is the first-order Taylor expansion to approximate a token's influence. This linear approximation is typically accurate only for very small changes. Given the highly non-linear nature of large multimodal models, what is the theoretical or empirical justification for assuming that this local gradient is a reliable proxy for a token's true, global contribution to the final output? Could you provide evidence to validate that this approximation holds true for complex semantic tasks like mitigating co-occurrence bias across an entire generated sequence?

---

### Official Review · Reviewer_9Vpf · 2025-11-03

**Soundness:** 3
**Presentation:** 3
**Contribution:** 2
**Rating:** 2
**Confidence:** 3

**Summary:**

This paper addresses hallucinations in MLLMs, where generated text is inconsistent with visual inputs. The authors identify two root causes: text-visual bias, an over-reliance on text prompts and generation history, and co-occurrence bias, which stems from spurious correlations between objects in training data. The main contribution is GACD, an inference-time method that does not require model finetuning or auxiliary models. GACD uses first-order Taylor gradients to estimate the influence of individual visual and text tokens on the output logits.  Experiments on benchmarks including AMBER, POPE, and LLaVA-QA90 show GACD significantly reduces hallucination rates and improves visual grounding.

**Strengths:**

1.  The method provides a principled bias estimation using first-order Taylor gradients to quantify token influence, avoiding heuristics and offering a clear mechanism for interpreting modal contributions (Section 3.2).
2.  The method is validated across a diverse set of MLLMs and benchmarks, including both generative and discriminative tasks, demonstrating its general applicability and consistent performance gains (Tables 1-4, 6).

**Weaknesses:**

### About Method

1.  One of the paper's core contributions is the adaptive weight `α_m`, but the derivation of Equation 12 is highly unclear and potentially erroneous in the main text, making it difficult for readers to understand how it is derived from the goal of "balancing cross-modal influence." The authors should provide a complete and rigorous mathematical derivation in the methodology section.
2.  In the "Object-aware Visual Token Grouping" component, the method assigns a single visual token with the maximum influence to each noun via `argmax`. This one-to-one hard assignment may be too restrictive, as a single object can correspond to multiple visual feature blocks. The paper should discuss the limitations of this design and ideally validate its rationale through experiments (e.g., by comparing it with top-k or threshold-based approaches).
3.  The paper proposes a sample-dependent early stopping criterion based on visual influence, yet its threshold `ε` is set empirically for different models. This weakens the method's "plug-and-play" nature. It is recommended that the authors investigate more principled ways to automatically determine this threshold, or at least analyze its sensitivity across different models and tasks.
4.  The proposed method relies on computing gradients and performing two forward passes at each decoding step, leading to a significant computational overhead (a 101% increase in TFLOPs as noted in Appendix R). While this is expected for decoding-time methods, the paper should discuss this cost more explicitly in the main experimental section and compare it with the computational costs of other baselines to allow for a comprehensive assessment of its practical feasibility.

### About Experiment

1.  The experiments indicate a slight performance degradation on certain complex reasoning tasks (e.g., sub-tasks in MMBench). It is recommended that the authors more explicitly discuss the method's applicability boundaries in the main paper and analyze why emphasizing visual grounding might affect abstract reasoning capabilities.
2.  A core innovation is the gradient-based "Object-aware Visual Token Grouping" mechanism, yet the experimental section lacks direct validation of its accuracy. It is suggested to add experiments, for instance, by using datasets with segmentation masks to quantitatively verify whether the selected tokens accurately correspond to the target objects, thereby enhancing the method's interpretability and reliability.
3.  The paper claims that the method "preserves information," but experimental results (e.g., Table 2) consistently show a slight decrease in recall. It is advisable for the authors to describe this phenomenon more precisely, discussing it as a favorable "accuracy-informativeness" trade-off rather than an absolute "preservation of information."
4.  Computational overhead is a crucial metric for evaluating inference-time methods, and the appendix reveals a >100% increase in TFLOPs. It is recommended to move this critical practical data to the main experimental section for reporting and discussion to improve transparency.

**Questions:**

1.  Could you provide a detailed derivation for $α_m$ in Equation 12? The formula does not seem to directly follow from the stated goal of "matching the influence of `t^u` with the dominant text-level influence $I_m^l$." A more direct derivation would seem to yield $α_m = I_m\^l / I_m^u - 1$. Please clarify the detailed derivation and final form of your method.

---

### Note · Authors · 2025-11-12

I have read and agree with the venue's withdrawal policy on behalf of myself and my co-authors.